# ABBA-ADAPTERS: EFFICIENT AND EXPRESSIVE FINE-TUNING OF FOUNDATION MODELS

**Raghav Singhal\*[1], Kaustubh Ponkshe\*[1], Rohit Vartak\*[2], Praneeth Vepakomma[1,3]**
[1]Mohamed bin Zayed University of Artificial Intelligence, [2]Duke University,
[3]Massachusetts Institute of Technology

## ABSTRACT

Large Language Models have demonstrated strong performance across a wide range of tasks, but adapting them efficiently to new domains remains a key challenge. Parameter-Efficient Fine-Tuning (PEFT) methods address this by introducing lightweight, trainable modules while keeping most pre-trained weights fixed. The prevailing approach, LoRA, models updates using a low-rank decomposition, but its expressivity is inherently constrained by the rank. Recent methods like HiRA aim to increase expressivity by incorporating a Hadamard product with the frozen weights, but still rely on the structure of the pre-trained model. We introduce **ABBA**, a new PEFT architecture that reparameterizes the update as a Hadamard product of two independently learnable low-rank matrices. In contrast to prior work, ABBA fully decouples the update from the pre-trained weights, enabling both components to be optimized freely. This leads to significantly higher expressivity under the same parameter budget, a property we validate through matrix reconstruction experiments. Empirically, ABBA achieves state-of-the-art results on arithmetic and commonsense reasoning benchmarks, consistently outperforming existing PEFT methods by a significant margin across multiple models. Our code is publicly available at: https://github.com/CERT-Lab/abba.

## 1 INTRODUCTION

Large Language Models (LLMs) have become the backbone of modern NLP systems (1–7), demonstrating strong generalization across a wide range of tasks (8; 9). However, adapting these models to new tasks typically requires full fine-tuning (FT), which is computationally and memory intensive. Parameter-Efficient Fine-Tuning (PEFT) methods address this challenge by introducing a small number of trainable parameters while keeping the majority of model weights frozen (10–12). Among PEFT approaches, Low-Rank Adaptation (LoRA) (10) is the most widely adopted due to its simplicity and effectiveness. It models the weight update $\Delta W$ as the product of two low-rank matrices, providing a compact parameterization. However, this formulation inherently constrains updates to a low-dimensional subspace, limiting expressivity. Several extensions attempt to overcome this limitation: LoRA-XS (12) restricts updates to a frozen, SVD-derived subspace and training only a small higher-rank intermediary matrix while DoRA (11) modifies only the directional component of the weights via low-rank updates. Despite these architectural variations, all retain LoRA's core constraint: updates remain strictly low-rank and limited in expressivity.

HiRA (13) addresses LoRA's limited expressivity by applying a Hadamard product between a low-rank update and the frozen pre-trained weights $W_0$, enabling updates that can, in principle, attain full rank. However, HiRA's expressivity remains tightly coupled to $W_0$, as the learned adapters merely modulate the pre-trained weights rather than generating the full update independently. For example, if the target update equals $\mathrm{diag}(W_0)$, HiRA must learn adapters that approximate the identity matrix, an orthonormal structure that is challenging for low-rank modules to represent accurately (Figure 2).

In this work, we introduce **ABBA**, a novel architecture framework that reparameterizes the weight update as the Hadamard product of two fully learnable low-rank matrices (see Figure 1). Each component is independently formed via a low-rank decomposition ($B_1 A_1$ and $B_2 A_2$), resulting

---

\* denotes equal contribution. Author order decided randomly.

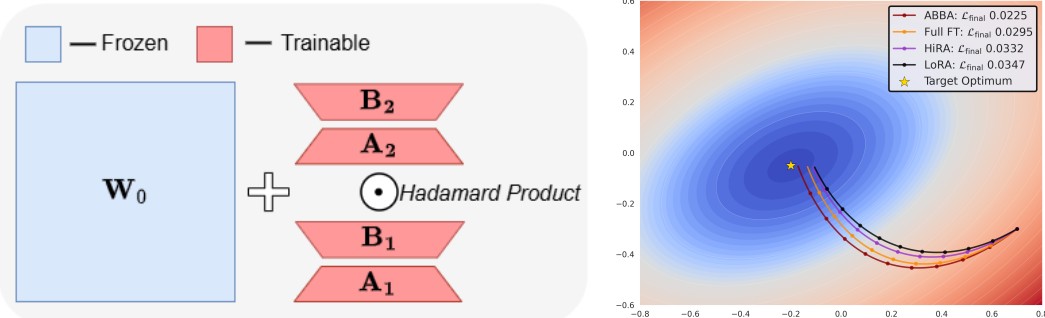

Figure 1: **Left:** Illustration of ABBA's parameterization, where the update is expressed as the Hadamard product of two learnable low-rank matrices. **Right:** A toy experiment demonstrating ABBA's optimization behavior. We first train a 2-layer MLP to classify the first 8 MNIST digits, then fine-tune it to recognize the last 2. ABBA converges faster and achieves better final performance.

in a highly expressive update while maintaining parameter counts. Unlike HiRA, ABBA is fully decoupled from the pretrained weights $W_0$, allowing both components to be optimized without structural constraints. The name ABBA reflects the four low-rank matrices that define the architecture.

We analyze expressivity through a matrix reconstruction task, where Hadamard-structured updates consistently outperform standard LoRA decompositions under the same parameter budget (Figure 2). This demonstrates that ABBA can represent a broader class of updates than LoRA within identical constraints. A toy MNIST experiment (Figure 1) shows that ABBA converges to a solution significantly closer to the true optimum compared to LoRA and HiRA, indicating that its increased expressivity is also practically accessible during learning. We present an exact reformulation of the ABBA update using the Khatri–Rao matrix factorization, enabling efficient implementation without approximation. Empirically, ABBA consistently outperforms existing PEFT methods across a broad range of tasks and models, all within the same or lower parameter budget. In addition, we conduct extensive ablation studies to validate the effectiveness of our design choices and hyperparameter settings. Our key contributions are summarized as:

- We propose ABBA, a novel PEFT architecture that models the weight update as the Hadamard product of two independently learnable low-rank matrices. This formulation enables highly expressive, high-rank updates while preserving strict parameter efficiency.
- We provide empirical analyses of ABBA's expressivity, showing that Hadamard-based decomposition consistently outperforms standard low-rank methods in matrix reconstruction.
- We introduce an exact and efficient reformulation of ABBA using Khatri–Rao factorization, enabling scalable and practical implementation without compromising expressivity.
- Through extensive experiments on four models across arithmetic and commonsense reasoning tasks, we demonstrate that ABBA achieves state-of-the-art performance, significantly outperforming existing PEFT methods under equal or lower parameter budgets.

## 2 METHODOLOGY

### 2.1 PRELIMINARIES

**Full Fine-Tuning.** Given a pre-trained weight matrix $W_0 \in \mathbb{R}^{m \times n}$, full FT updates all parameters via $W = W_0 + \Delta W$, introducing $m \times n$ trainable parameters per layer. This quickly becomes impractical due to the high memory and compute overhead.

**LoRA (10).** LoRA mitigates this by modeling the update as a low-rank decomposition: $\Delta W = sBA$, where $B \in \mathbb{R}^{m \times r}$, $A \in \mathbb{R}^{r \times n}$, and $s$ is a scaling factor. This reduces the number of trainable parameters to $r(m + n)$, with $r \ll \min(m, n)$. LoRA can represent any update of rank at most $r$, but cannot express higher-rank updates. Moreover, the projected gradient onto the weight space is also low-rank. While effective for simpler tasks, this limitation becomes significant in settings requiring high-rank updates or gradients (14; 15).

**HiRA (Hadamard High-Rank Adaptation) (13).** HiRA lifts LoRA's rank limitation by modulating its low-rank update with an element-wise (Hadamard) product with the frozen pre-trained weight $W_0$:

$$\Delta W = W_0 \odot (BA), \quad \text{where } \odot \text{ denotes the Hadamard product.} \tag{1}$$

This leverages the property that the Hadamard product of two matrices $W_1$ and $W_2$ with ranks $r_1$ and $r_2$ respectively satisfies $\text{rank}(W_1 \odot W_2) \leq r_1 \cdot r_2$. Thus, HiRA can produce updates of rank up to $r_0 r$, where $r_0 = \text{rank}(W_0)$, potentially addressing the low-rank limitation of LoRA. Additionally, the gradient projected onto $W$ is no longer low-rank. However, higher rank does not necessarily imply greater expressivity. Because HiRA's update is element-wise tied to $W_0$, it is restricted to a subspace defined by the pre-trained weights. This dependence can hinder generalization, especially in out-of-domain scenarios. As shown in Section 2.4, HiRA reduces reconstruction error over LoRA only when the element-wise ratio of the oracle update to $W_0$ is itself low-rank.

## 2.2 Improving the Expressivity of HiRA

A natural way to overcome HiRA's expressivity constraint is to make the matrix $W_h$ learnable:

$$\Delta W = W_h \odot (BA), \qquad \text{where } W_h \in \mathbb{R}^{m \times n} \text{ is trainable.} \tag{2}$$

However, this reintroduces the full $m \times n$ parameter cost of $W_h$, negating LoRA's core efficiency advantage. Even if $W_h$ is fixed but not equal to $W_0$, the additional memory required to store it significantly increases the overhead. In contrast, HiRA sets $W_h = W_0$, which is already stored, thereby preserving LoRA's parameter and memory efficiency. This raises a critical question:

***Can we achieve greater expressivity and high-rank learning while maintaining the parameter and memory efficiency of LoRA?***

Importantly, we note that full-rank updates are not always necessary. Moreover, $W_h$ itself does not need to be full-rank. Since any $m \times n$ matrix has rank at most $r_0 = \min(m, n)$, a modulation matrix with rank above $r_0/r$ offers no additional expressivity for the Hadamard product.

## 2.3 ABBA-Adapters: Improved Expressivity and High Rank, Yet Efficient

As discussed above, high-rank updates do not require $W_h$ to be full-rank. Leveraging this insight, we reparameterize $W_h$ as the Hadamard product of two independently learnable low-rank matrices, resulting in the following formulation:

$$\Delta W = s(B_1 A_1) \odot (B_2 A_2), \tag{3}$$

where $B_1 \in \mathbb{R}^{m \times r_1}$, $A_1 \in \mathbb{R}^{r_1 \times n}$ and $B_2 \in \mathbb{R}^{m \times r_2}$, $A_2 \in \mathbb{R}^{r_2 \times n}$, with $r_1, r_2 \ll \min(m, n)$ and $s$ is a scaling factor for stability. This parameterization introduces only $(r_1 + r_2)(m + n)$ parameters, significantly fewer than full FT, and achieves an effective rank up to $r_1 r_2$. To maximize expressivity under a fixed parameter budget, we set $r_1 = r_2$, as further supported by empirical results in Section 4.2. This preserves HiRA's ability to produce high-rank updates while improving expressivity, since all four matrices are independently learned. For fair comparison with LoRA and other PEFT baselines, we match parameter counts by setting $r_1 = r_2 = r/2$, so that ABBA and other methods use equivalent parameter budgets.

**Initialization of ABBA Adapters.** HiRA fixes the modulation matrix as $W_h = W_0$, directly tying the update to the pretrained weights. In contrast, **ABBA** makes this matrix fully learnable by reparameterizing it as $W_h = B_1 A_1$. We initialize the first adapter pair $(B_1, A_1)$ using the top-$r_1$ components from a truncated SVD of $W_0$, and the second pair $(B_2, A_2)$ using the standard LoRA initialization: $B_2$ as zeros and $A_2$ with Kaiming uniform sampling.

$$U_{r_1}, \Sigma_{r_1}, V_{r_1}^\top \leftarrow \textbf{SVD}_{r_1}(W_0), \tag{4}$$

$$B_1 \leftarrow U_{r_1} \Sigma_{r_1}^{1/2}, \quad A_1 \leftarrow \Sigma_{r_1}^{1/2} V_{r_1}^\top, \quad B_2 \leftarrow \mathbf{0}, \quad A_2 \leftarrow \mathcal{N}(0, \sigma^2). \tag{5}$$

By the Eckart–Young–Mirsky (EYM) theorem (16; 17), the truncated SVD yields the optimal rank-$r_1$ approximation of $W_0$. This hybrid initialization anchors the update close to a meaningful low-rank subspace, while enabling the second adapter pair to explore task-specific directions during training. We validate the effectiveness of this strategy empirically in Section 4.1.

**Making ABBA Memory-Efficient.** While ABBA is clearly parameter-efficient, analyzing its memory footprint during training is more subtle. In LoRA, the update $\Delta W = BA$ is applied as $\Delta W x = B(Ax)$, allowing intermediate computations to remain low-rank. Only the activation $Ax \in \mathbb{R}^r$ and the adapter weights need to be stored additionally, avoiding the materialization of the full $m \times n$ matrix $BA$. In contrast, ABBA's update $\Delta W = (B_1 A_1) \odot (B_2 A_2)$ poses a challenge. A naive implementation would require constructing both $B_1 A_1$ and $B_2 A_2$, followed by their elementwise product, resulting in the storage of multiple full $m \times n$ matrices. Moreover, unlike LoRA, the Hadamard product does not distribute over matrix–vector multiplication, so computing $B_2(A_2 x)$ does not help incorporate the other matrices.

> **Theorem 1** (Khatri–Rao Factorization (18)). *Let $B_1 A_1, B_2 A_2 \in \mathbb{R}^{m \times n}$. Then,* $(B_1 A_1) \odot (B_2 A_2) = \underbrace{(B_1 \odot_r B_2)}_{m \times r_1 r_2} \underbrace{(A_1^\top \odot_r A_2^\top)^\top}_{r_1 r_2 \times n}$, *where $\odot_r$[1] denotes the row-wise Khatri–Rao product.*

To address this, we use Theorem 1 to rewrite ABBA in a LoRA-like form: let $B_{\mathrm{kr}} = B_1 \odot_r B_2$ and $A_{\mathrm{kr}} = (A_1^\top \odot_r A_2^\top)^\top$. The update becomes $\Delta W x = B_{\mathrm{kr}}(A_{\mathrm{kr}} x)$, avoiding any full-rank construction.

This enables ABBA to match LoRA's compute and memory efficiency, while offering significantly higher expressivity, and remain more efficient than variants like HiRA, as shown in Section 4.4.

## 2.4 EXPRESSIVITY OF ABBA

The expressivity of a matrix reparameterization can be evaluated by its ability to accurately reconstruct arbitrary target matrices, relative to alternative parameterizations.

**LoRA and HiRA.** In LoRA, the weight update is modeled as a low-rank decomposition $\Delta W = BA$. For any matrix $M \in \mathbb{R}^{m \times n}$, the reconstruction error of this approximation is defined as:

$$\mathcal{E}(r) = \|M - BA\|_F, \tag{6}$$

and is lower-bounded by the classical EYM theorem (16; 17), which states that the optimal rank-$r$ approximation is given by the truncated SVD. Since a LoRA adapter of rank $r$ can only represent updates with rank at most $r$, EYM provides a theoretical minimum for $\mathcal{E}(r)$. LoRA achieves this bound exactly when the learned adapters align with the top singular components of $M$; otherwise, practical considerations such as suboptimal initialization may lead to a performance gap.

A similar bound can be derived for HiRA when the modulation matrix $W_0$ has all nonzero entries. In this case, the optimal Hadamard-structured approximation can be obtained by element-wise dividing $W$ by $W_0$, followed by applying truncated SVD to the resulting matrix. The expressivity advantage of HiRA over LoRA arises only if $\mathrm{rank}(\Delta W / W_0) < \mathrm{rank}(\Delta W)$. Otherwise, for a general update $\Delta W$, HiRA has the same reconstruction error bound as LoRA, as characterized by the EYM theorem.

**ABBA vs. LoRA: Reconstruction.** Unlike SVD-based methods, ABBA does not admit a closed-form solution for its low-rank factors (see Appendix B for understanding why). We thus evaluate its expressivity by comparing the reconstruction error versus other methods. Given a target matrix $M \in \mathbb{R}^{m \times n}$, we define the reconstruction error for a method $X \in \{\mathrm{LoRA}, \mathrm{ABBA}\}$ at rank $r$ as: $E_{X,r} = \min_{M_{X,r}} \|M - M_{X,r}\|_F^2$, where the LoRA approximation is the truncated SVD $M_{\mathrm{SVD},r} = U\Sigma_r V^\top$, and the ABBA approximation is given by $M_{\mathrm{ABBA},r} = (B_1 A_1) \odot (B_2 A_2)$.

Prior work (19) establishes a loose upper bound $E_{\mathrm{ABBA},r} \leq E_{\mathrm{LoRA},r}$, which holds trivially by setting one ABBA factor to the rank-$r$ SVD and the other to an all-ones matrix. Empirically, however, they observe a stronger trend: $E_{\mathrm{ABBA},r} \lesssim E_{\mathrm{LoRA},2r}$, which suggests that ABBA can match or outperform a rank-$2r$ SVD approximation using only rank $r$. However, this is not guaranteed for arbitrary matrices, as the quality of reconstruction depends on the spectral properties and structure of the matrix. The only known theoretical comparison between the two is the bound: $E_{\mathrm{LoRA},2r} - E_{\mathrm{ABBA},r} \leq \sum_{i=2r+1}^{r^2} \sigma_i^2$, where $\sigma_i$ are the singular values of $M$ (19). While this bound offers some insight, it is loose and does not guarantee a strict ordering between the reconstruction errors.

---

[1] Given $U, V \in \mathbb{R}^{m \times n}$, the row-wise Khatri–Rao product $U \odot_r V \in \mathbb{R}^{m \times n^2}$ is defined by $[U \odot_r V]_i := [U_{i1} V_i, U_{i2} V_i, \ldots, U_{in} V_i]$, where $V_i$ is the $i$-th row of $V$.

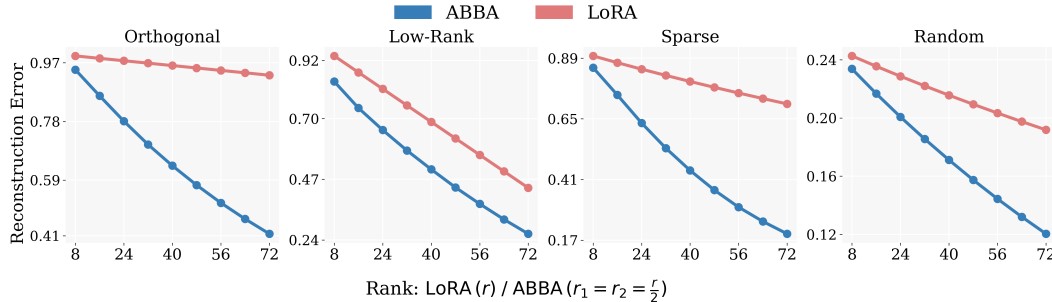

Figure 2: **Empirical Reconstruction Errors.** We compare ABBA and LoRA decompositions across various matrix types by measuring reconstruction error $\mathcal{E}(r)$ under equal parameter budgets. For each LoRA rank $r$, we set ABBA ranks to $r_1 = r_2 = r/2$ for a fair comparison. ABBA consistently achieves significantly lower reconstruction error than LoRA, across all matrix types.

To better understand practical behavior, we empirically evaluate the reconstruction error of different parameterizations across diverse matrix types. As shown in Figure 2, the ABBA-based Hadamard reparameterization consistently achieves lower reconstruction error than standard LoRA, indicating greater expressivity. This aligns very well with prior work leveraging Hadamard structures for efficient and expressive matrix representations (20), further validating our formulation.

## 2.5 STABILITY OF THE ABBA UPDATE

The scaling factor $s$ is critical in controlling the optimization dynamics of the ABBA update. It must be chosen carefully to scale appropriately with the ranks $r_1$ and $r_2$ of the underlying low-rank factors. If set too low, learning stagnates; if too high, training may diverge. While ABBA resembles LoRA in structure, its effective rank is $r_1 r_2$, and the scaling behavior must reflect this increased capacity. Inspired by the scaling analysis in rsLoRA (21), where stability is achieved under a complexity of $\mathcal{O}(1)$ with respect to rank, one might expect similar scaling for ABBA. However, since ABBA and LoRA inhabit different parameter spaces, these arguments do not transfer directly. To formalize this, we introduce the notion of *rank-stability* for ABBA in Definition 1, which ensures that forward and backward dynamics remain well-conditioned as $r_1$ and $r_2$ vary.

**Definition 1** (Rank Stability of ABBA Adapters (21; 22)). *An ABBA adapter of the form $s_{ABBA}(B_1 A_1) \odot (B_2 A_2)$ is **rank-stabilized** if the following conditions hold:*

1. *If the 2$^{nd}$ moment of the input is $\Theta_{r_1,r_2}(1)$ in each entry with inputs being i.i.d, then the 2$^{nd}$ moment of the outputs of the adapter is also $\Theta_{r_1,r_2}(1)$ in each entry.*

2. *If the 2$^{nd}$ moment of the loss gradient with respect to the adapter outputs is $\Theta_{r_1,r_2}(1)$ in each entry; then the 2$^{nd}$ moment of the loss gradient of the input of the adapter is also $\Theta_{r_1,r_2}(1)$ in each entry.*

Building on Definition 1, we establish in Theorem 2 that the ABBA update is indeed rank-stable. We empirically demonstrate that our scaling yields improved stability, as evidenced by the gradient norm experiments shown in Figure 5 in Appendix I.

**Theorem 2** (Rank-Stability of ABBA). *Let the ABBA update be $\Delta W = s_{ABBA}(B_1 A_1) \odot (B_2 A_2)$, with $B_1, A_1, B_2, A_2$ independent, mean-zero random matrices with finite variance. Then, the update satisfies the stability conditions of Definition 1 if and only if:*

$$s_{ABBA} \in \Theta\left(\frac{1}{\sqrt{r_1 r_2}}\right).$$

*Proof.* See Appendix D. □

**Final ABBA Parameterization.** We express the ABBA scaling factor as $s_{\text{ABBA}} = \frac{\alpha^2}{\sqrt{r_1 r_2}}$, since Theorem 2 shows that $s_{\text{ABBA}} \in \Theta\left(\frac{1}{\sqrt{r_1 r_2}}\right)$. With this choice, the ABBA update is:

$$\Delta W = \frac{\alpha^2}{\sqrt{r_1 r_2}} (B_1 A_1) \odot (B_2 A_2). \tag{7}$$

Equivalently, to highlight the connection with LoRA, ABBA can be expressed as the Hadamard product of two rank-stabilized LoRA adapters:

$$\Delta W = \left(\frac{\alpha}{\sqrt{r_1}} B_1 A_1\right) \odot \left(\frac{\alpha}{\sqrt{r_2}} B_2 A_2\right). \tag{8}$$

## 3 EXPERIMENTS

We evaluate ABBA on a range of models, specifically Llama-3.2 1B (23), Llama-3.2 3B (23), Mistral-7B (24), and Gemma-2 9B (25), to test its effectiveness across diverse scales and architectures. We run experiments on multiple benchmarks to capture varied trends. Appendix L details our training configurations, and Appendix M lists dataset specifics. For fair comparison with LoRA and other PEFT methods, we match the number of trainable parameters in ABBA by setting $r_1 = r_2 = r/2$.

**Baselines.** We compare ABBA against full fine-tuning, LoRA (10), and several strong LoRA variants: rsLoRA (21), PiSSA (26), DoRA (11), LoRA-Pro (14), and HiRA (13).

### 3.1 COMMONSENSE REASONING

We fine-tune Llama-3.2 models at 1B and 3B scales (23) on COMMONSENSE170K, a multi-task dataset comprising eight commonsense reasoning benchmarks (27). These include OBQA (28), ARC-Challenge and ARC-Easy (29), WinoGrande (30), HellaSwag (31), PIQA (32), SIQA (33), and BoolQ (34). We evaluate performance on each dataset independently to capture task-specific generalization. We insert LoRA modules into the key, query, and value projections, the attention output, and all feedforward layers. Table 1 reports the results. ABBA consistently outperforms all other PEFT methods across both models, and in many cases surpasses full FT.

Table 1: Comparison of multiple FT methods on Llama-3.2 1B and 3B across eight commonsense reasoning datasets. Best results among PEFT methods are in **bold**.

| Model | Method | # Params | OBQA | ARC-c | ARC-e | Wino | HellaS | PIQA | SIQA | BoolQ | Avg. |
|---|---|---|---|---|---|---|---|---|---|---|---|
| | | | Accuracy (↑) | | | | | | | | |
| Llama-3.2 1B | Full FT | 1.24 B | 73.42 | 63.70 | 78.88 | 75.12 | 80.98 | 80.22 | 74.79 | 66.21 | 74.17 |
| | LoRA | 22.54 M | 71.83 | 59.13 | 74.32 | 73.87 | 74.96 | 78.13 | 73.75 | 65.96 | 71.49 |
| | rsLoRA | 22.54 M | 71.11 | 59.85 | 74.90 | 73.85 | 75.34 | 78.32 | 73.47 | 65.44 | 71.54 |
| | PiSSA | 22.54 M | 71.45 | 60.32 | 74.43 | 72.90 | 75.65 | 78.45 | 73.63 | 65.83 | 71.58 |
| | DoRA | 22.92 M | 71.99 | 60.98 | 77.65 | 73.42 | 76.33 | 78.81 | 73.79 | 65.91 | 72.36 |
| | LoRA-Pro | 22.54 M | 71.68 | 61.11 | 76.37 | 73.12 | 76.89 | 79.24 | 74.02 | 65.79 | 72.28 |
| | HiRA | 22.54 M | 72.18 | 61.26 | 78.37 | 72.06 | 78.87 | 79.59 | 74.41 | 65.32 | 72.76 |
| | ABBA$_{r=16}$ | 11.27 M | 71.86 | 63.05 | 78.33 | 73.95 | 80.93 | **80.63** | **75.33** | 65.96 | 73.76 |
| | ABBA$_{r=32}$ | 22.54 M | **75.06** | **64.59** | **79.74** | **76.03** | **82.50** | 80.41 | 75.08 | **66.80** | **75.03** |
| Llama-3.2 3B | Full FT | 3.21 B | 81.88 | 75.29 | 88.52 | 85.02 | 91.92 | 85.64 | 80.45 | 70.43 | 82.39 |
| | LoRA | 48.63 M | 81.87 | 74.32 | 86.91 | 82.24 | 90.71 | 85.20 | 79.12 | 70.03 | 81.30 |
| | rsLoRA | 48.63 M | 81.72 | 74.18 | 86.71 | 82.02 | 90.45 | 85.05 | 78.92 | 69.81 | 81.11 |
| | PiSSA | 48.63 M | 81.79 | 74.61 | 87.23 | 82.68 | 90.88 | 85.42 | 79.44 | 70.12 | 81.52 |
| | DoRA | 49.40 M | 82.04 | 74.87 | 87.61 | 82.90 | 90.76 | 85.63 | 79.68 | 70.43 | 81.74 |
| | LoRA-Pro | 48.63 M | 81.74 | 75.32 | 87.24 | 83.42 | 90.90 | 85.81 | 79.35 | 71.28 | 81.88 |
| | HiRA | 48.63 M | 81.58 | 76.38 | 88.76 | 83.95 | 91.67 | 85.61 | 79.91 | 72.69 | 82.56 |
| | ABBA$_{r=16}$ | 24.32 M | 83.40 | 77.39 | 89.56 | 85.16 | 93.51 | **86.89** | 80.55 | 73.03 | 83.68 |
| | ABBA$_{r=32}$ | 48.63 M | **85.04** | **79.10** | **89.61** | **85.24** | **92.37** | 86.83 | **80.96** | **73.52** | **84.08** |

## 3.2 ARITHMETIC REASONING

We fine-tune Mistral-7B (24) and Gemma-2 9B (25) on a 20K-sample subset of MetaMathQA (35), and evaluate their performance on GSM8K (36) and MATH (37). We insert LoRA adapters into all attention projections (query, key, value, and output) as well as both feedforward layers. We report results in Table 2. ABBA achieves superior performance over all other PEFT approaches across both models, and often outperforms full FT. We hypothesize that this effect arises because ABBA's structured parameterization implicitly regularizes training and enables more efficient task-specific adaptation than unconstrained full FT.

Table 2: Comparison of multiple FT methods on Mistral-7B and Gemma-2 9B across arithmetic reasoning benchmarks. Best results among PEFT methods are in **bold**.

| Method | Mistral-7B | | | Gemma-2 9B | | |
|---|---|---|---|---|---|---|
| | # Params | GSM8K ($\uparrow$) | MATH ($\uparrow$) | # Params | GSM8K ($\uparrow$) | MATH ($\uparrow$) |
| Full FT | 7.24 B | 63.87 | 17.65 | 9.24 B | 79.23 | 38.02 |
| LoRA | 83.88 M | 61.94 | 15.98 | 108.04 M | 76.19 | 36.56 |
| rsLoRA | 83.88 M | 62.15 | 16.24 | 108.04 M | 76.84 | 36.88 |
| PiSSA | 83.88 M | 62.43 | 16.52 | 108.04 M | 77.12 | 37.04 |
| DoRA | 85.26 M | 62.65 | 16.64 | 109.88 M | 77.58 | 37.04 |
| LoRA-Pro | 83.88 M | 63.07 | 17.32 | 108.04 M | 78.26 | 37.53 |
| HiRA | 83.88 M | 63.15 | 17.44 | 108.04 M | 78.47 | 38.22 |
| ABBA$_{r=16}$ | 41.94 M | 64.97 | 18.06 | 54.02 M | 78.70 | 38.41 |
| ABBA$_{r=32}$ | 83.88 M | **66.26** | **18.08** | 108.04 M | **79.76** | **39.18** |

## 4 ANALYSIS

### 4.1 INITIALIZATION STRATEGIES FOR ABBA

Initialization of the adapter matrices $B_1, A_1$ and $B_2, A_2$ is crucial to ABBA's performance. A naive LoRA-style initialization, where $B_1, B_2$ are set to zero and $A_1, A_2$ use Kaiming uniform, leads to training failure due to zeroed-out gradients (see Appendix C). To address this, we explore several initialization strategies that combine truncated SVD-based approximations with standard schemes, summarized in Table 3.

Inspired by PiSSA-LoRA (26), one approach is to approximate the base weight $W_0$ at initialization. This can be done by initializing one adapter pair using the truncated SVD of $W_0$, and the other with scaled constant values (e.g., ones) to prevent gradient explosion **(1)**. Another strategy initializes both adapter pairs using the top-$r/2$ components from the truncated SVD of $\sqrt{W_0}$ **(2)**. A variation of this approach assigns the top-$r/2$ components to one adapter pair and the next-$r/2$ to the other, introducing greater representational diversity and yielding slightly improved results **(3, 4)**. We also consider a hybrid strategy, where one adapter pair approximates $W_0$ via truncated SVD, and the other follows LoRA-style initialization (Kaiming for $A$, zeros for $B$). This configuration performs best and closely resembles the initialization used in HiRA **(5)**.

Our final method adopts this approach: $B_1, A_1$ are initialized using the truncated SVD of $W_0$, while $B_2, A_2$ follow LoRA-style initialization **(Ours)**.

Table 3: Comparison of different initialization strategies for ABBA (Mistral-7B).

| Initialization Method | GSM8K | MATH |
|---|---|---|
| **(1)** $B_1, A_1 \leftarrow$ top-$r/2$ from Trunc. SVD($W_0$), $B_2, A_2 \leftarrow$ Ones (scaled) | 57.39 | 10.88 |
| **(2)** Both adapters: top-$r/2$ from Trunc. SVD($\sqrt{W_0}$) | 64.53 | 16.57 |
| **(3)** First adapter: top-$r/2$, second: next-$r/2$ from Trunc. SVD($\sqrt{W_0}$) | 64.86 | 17.05 |
| **(4)** Second adapter: top-$r/2$, first: next-$r/2$ from Trunc. SVD($\sqrt{W_0}$) | 64.79 | 17.16 |
| **(5)** $B_2, A_2 \leftarrow$ top-$r/2$ from Trunc. SVD($W_0$), $B_1, A_1 \leftarrow$ LoRA Init (Zeros, Kaiming) | 66.19 | 18.06 |
| **Ours:** $B_1, A_1 \leftarrow$ top-$r/2$ from Trunc. SVD($W_0$), $B_2, A_2 \leftarrow$ LoRA Init (Zeros, Kaiming) | **66.26** | **18.08** |

## 4.2 CHOOSING IMPORTANT HYPERPARAMETERS

**Selecting $\alpha$.** To empirically validate this, we sweep over a range of $\alpha$ values for Llama-3.2 3B, in Table 4. Consistent with prior PEFT literature, ABBA achieves optimal performance within the typical LoRA scaling range of $16 - 32$. Additional evidence is provided in Table 8 (Appendix E).

Table 4: Performance comparison across different $\alpha$ values for Llama-3.2 3B.

| $\alpha$ | Accuracy ($\uparrow$) | | | | | | | | |
|---|---|---|---|---|---|---|---|---|---|
| | **BoolQ** | **PIQA** | **SIQA** | **HellaS.** | **WinoG.** | **ARC-e** | **ARC-c** | **OBQA** | **Avg.** |
| 4 | 70.92 | 85.74 | 79.84 | 92.07 | 84.45 | 88.38 | 75.83 | 82.20 | 82.43 |
| 8 | 71.19 | 86.83 | 80.65 | 92.60 | 85.95 | 88.47 | 76.11 | 82.20 | 82.97 |
| 12 | 72.35 | 86.62 | 81.63 | 92.82 | 85.01 | 89.10 | 77.05 | 83.00 | 83.45 |
| 16 | 72.88 | 86.45 | 80.75 | 93.18 | 86.97 | 89.98 | 78.33 | 83.80 | 84.04 |
| 24 | 73.82 | 85.91 | 80.55 | 93.29 | 85.87 | 89.64 | 78.41 | 84.60 | 84.01 |
| 32 | 73.52 | 86.93 | 80.96 | 92.73 | 85.24 | 89.61 | 79.10 | 85.04 | 84.08 |
| 48 | 71.83 | 84.77 | 78.96 | 90.52 | 84.92 | 87.12 | 74.57 | 82.40 | 81.88 |
| 64 | 67.71 | 79.43 | 77.38 | 81.25 | 78.69 | 79.96 | 66.47 | 80.00 | 76.36 |

**Selecting $r_1, r_2$.** An important choice is allocating the total rank budget $r = r_1 + r_2$ between the two low-rank projections. A balanced setting, $r_1 = r_2 = r/2$, is expected to perform best since it maximizes the effective rank $r_1 r_2$, increasing expressivity. We empirically evaluate various $\{r_1, r_2\}$ combinations under a fixed total rank on Mistral-7B in Table 5. The symmetric configuration achieves the best accuracy, consistent with our hypothesis.

Table 5: Different $\{r_1, r_2\}$ pairs with fixed $r_1 + r_2 = 32$.

| $r_1$ | $r_2$ | GSM8K | MATH |
|---|---|---|---|
| 4 | 28 | 64.43 | 17.01 |
| 8 | 24 | 63.91 | 17.20 |
| 12 | 20 | 64.29 | 18.22 |
| 16 | 16 | 66.26 | 18.08 |

## 4.3 PLACEMENT OF ABBA IN TRANSFORMERS

Figure 3 examines the effect of fine-tuning individual transformer components in ABBA, namely Query, Key, Value, Output, Up, Gate, and Down projections. The results show the following: Query/Key contribute the least, followed by Value/Up, while Gate/Output/Down are the most impactful. This reflects their functional roles: Query/Key support attention scoring, whereas the others play a more direct role in transforming and retaining learned representations.

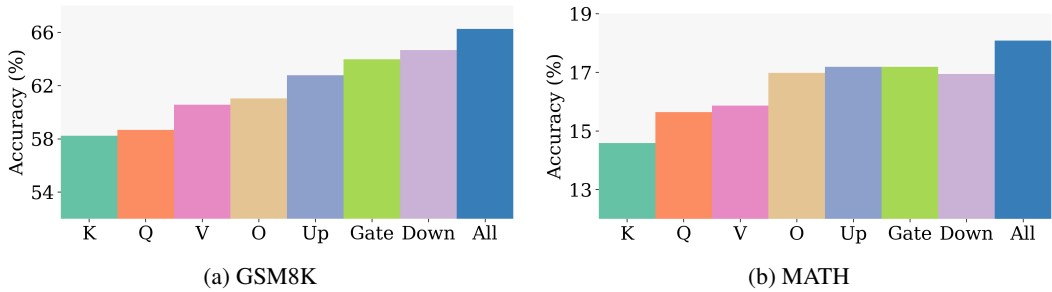

(a) GSM8K      (b) MATH

Figure 3: Impact of selectively fine-tuning individual transformer components - Key, Query, Value, Output, Up, Gate, and Down projections, with ABBA (Mistral-7B).

## 4.4 ALL ABOUT EFFICIENCY

**Training Memory Footprint.** We report peak memory usage for various methods in Figure 4, measured with a batch size of 1 and a context length of 256. ABBA reduces memory consumption by $\approx 3 - 3.5$ times compared to full FT. Compared to other PEFT methods, ABBA offers similar memory efficiency to LoRA, and is $\approx 30 - 35\%$ more efficient than the next-best method, HiRA.

**Training Time.** We benchmark training time across multiple settings in Table 10 (Appendix G). ABBA has comparable training time to LoRA, with only a $\approx 2 - 3\%$ overhead, primarily due to the additional computation introduced by the Hadamard product. We clarify that the initialization

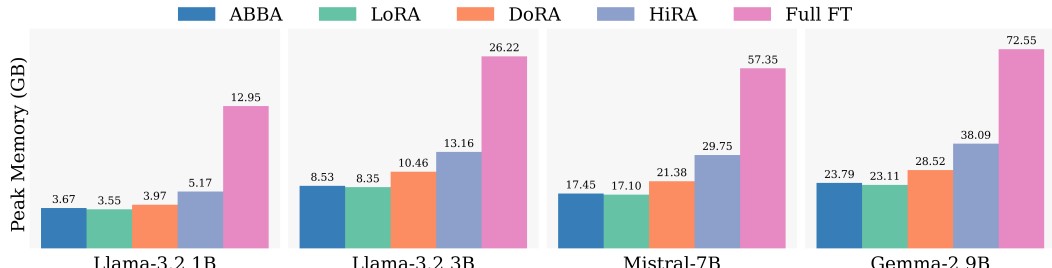

Figure 4: Comparison of training memory requirements across various methods. Results are reported for all models used in our work, with sequence length and batch size fixed at 256 and 1, respectively.

itself is highly efficient, as we compute the **truncated SVD** using `torch.svd_lowrank`. This initialization step takes less than one second for the entire model, even for the largest LLMs used in our experiments.

**Efficient Inference.**   At deployment time, ABBA supports efficient inference by pre-computing the update and merging it into the base weights as $W' = W_0 + (B_1 A_1) \odot (B_2 A_2)$. This allows models to switch tasks quickly by subtracting the update to restore $W_0$, then applying a new ABBA adapter. Since the update is fused ahead of inference, this incurs no runtime overhead or latency.

## 4.5   ADDITIONAL RESULTS ON ARITHMETIC TASKS

We report results for fine-tuning Mistral-7B and Gemma-2 9B using a larger 40K subset of Meta-MathQA, compared to the 20K subset used in the main experiments.

As shown in Table 6, ABBA continues to outperform all other PEFT baselines under matched parameter budgets. However, we observe that the performance gap between full fine-tuning and ABBA narrows when using the larger subset. This suggests that in higher data regimes, full fine-tuning may be a more suitable choice relative to other PEFT methods, provided the additional memory and compute requirements for full fine-tuning can be met.

Table 6: Comparison of multiple FT methods on Mistral-7B and Gemma-2 9B across arithmetic reasoning benchmarks (when finetuned on a 40K subset of MetaMathQA). Best results among PEFT methods are in **bold**.

| Method | Mistral-7B | | | Gemma-2 9B | | |
|---|---|---|---|---|---|---|
| | # Params | GSM8K (↑) | MATH (↑) | # Params | GSM8K (↑) | MATH (↑) |
| Full FT | 7.24 B | 66.28 | 18.34 | 9.24 B | 79.89 | 39.44 |
| LoRA | 83.88 M | 62.89 | 16.45 | 108.04 M | 77.02 | 37.05 |
| DoRA | 85.26 M | 63.24 | 16.67 | 109.88 M | 77.46 | 37.40 |
| HiRA | 83.88 M | 64.54 | 17.88 | 108.04 M | 78.90 | 38.53 |
| ABBA$_{r=16}$ | 41.94 M | 66.36 | 18.51 | 54.02 M | 79.93 | 39.27 |
| ABBA$_{r=32}$ | 83.88 M | **67.04** | **18.76** | 108.04 M | **80.31** | **39.92** |

## 4.6   EXTENDING ABBA VIA ADAPTER CHAINS

A natural extension of ABBA is decomposing the update into a composition of $k$ multiple adapter pairs: $\Delta W = B_1 A_1 \odot B_2 A_2 \cdots \odot B_k A_k$, where each adapter pair has rank $r/k$, with $r$ denoting the total rank budget. This factorized form increases the expressive capacity and the effective rank of the update. However, this introduces additional optimization challenges and may reduce training

Table 7: Standard ABBA (2 pairs) versus a chained extension using 4 pairs.

| Configuration | GSM8K | MATH |
|---|---|---|
| ABBA (2 pairs) | 66.26 | 18.08 |
| Chained ABBA (4 pairs) | 64.84 | 17.74 |

stability. We evaluate a variant of ABBA using a chain of four adapter pairs and compare it to the standard two-pair setup. As shown in Table 7, the chained version performs slightly worse, likely

due to suboptimal scaling or training instability. We show gradient norm plots of these variations in Figure 6 in Appendix J. We leave further investigation of multi-stage compositions to future work.

## 5 CONCLUSION

ABBA introduces a simple yet effective extension to the PEFT framework by expressing the weight update as a Hadamard product of two independently learnable low-rank matrices. This formulation retains the parameter efficiency of LoRA-style methods while enabling significantly higher expressivity. Our empirical results demonstrate that this expressivity translates to consistent performance gains across a range of tasks and models. Through a mathematically exact Khatri–Rao reformulation, ABBA matches LoRA's efficiency while representing high-rank updates entirely through low-rank components, offering a more powerful and efficient alternative to existing PEFT approaches.

## REPRODUCIBILITY STATEMENT

We have made considerable efforts to ensure that our work is fully reproducible. Our code is open-source and available at: https://github.com/CERT-Lab/abba. Detailed descriptions of our experimental settings can be found in Section 3 and Appendix L. All datasets in this work are widely used, publicly available benchmarks (see Appendix M for details).

## ACKNOWLEDGEMENTS AND DISCLOSURE OF FUNDING

This research was supported by Mohamed bin Zayed University of Artificial Intelligence (MBZUAI), with partial funding from the ADIA Lab Fellowship.

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

# Appendix

CONTENTS

## A    RELATED WORK

**Parameter-Efficient Fine-Tuning (PEFT) and Low-Rank Adaptation (LoRA).** PEFT methods adapt large pretrained models to downstream tasks by training a small number of additional parameters while keeping the base model frozen. Among these, LoRA (10) is widely adopted for its simplicity and effectiveness, modeling the update $\Delta W$ as a low-rank product $BA$, thereby reducing trainable parameters significantly. Several extensions enhance LoRA along different axes. QLoRA (38) and QA-LoRA (39) combines quantization with LoRA to reduce memory footprint; AdaLoRA (40) allocates rank budgets dynamically across layers. LoRA-XS (12) inserts a small trainable core between frozen adapters to improve compression, while LoRA-Pro (14) and LoRA-SB (15) optimize adapters to better approximate full fine-tuning gradients. Other variants modify structure or initialization. VeRA (41) reuses frozen adapters across layers with task-specific scaling vectors; DoRA (11) applies low-rank updates only to the direction of pretrained weights. PiSSA (26) initializes adapters using top singular vectors, while rsLoRA (21) proposes scale-aware initialization to improve stability. LoRA-One (42) initializes adapters using this singular subspaces of the full fine-tuning gradient and incorporates preconditioning to improve convergence. WeGeFT (43) learns a small low-rank transformation that maps pretrained weights to their fine-tuned updates, either shared across layers or learned per layer. Despite all these differences, these methods share a core

principle: they represent $\Delta W$ using a low-rank structure, enabling adaptation under tight compute and memory constraints. LoRA-based methods have also been applied in other domains, such as federated fine-tuning (44; 20; 45; 46).

**Beyond Low-Rank: High-Rank and Structured Adaptation.** While low-rank PEFT methods offer strong efficiency–performance tradeoffs, they can underperform in tasks requiring high-rank updates. This limitation has driven recent efforts to move beyond purely low-rank parameterizations. HiRA (13) achieves high-rank updates by taking the Hadamard (elementwise) product of the pre-trained matrix $W_0$ with a low-rank adapter $BA$, leveraging the fact that such a product can increase effective rank without increasing parameter count. MoRA (47) instead learns a full-rank update through input compression and activation decompression, while KronA (48) uses Kronecker products between adapters to boost representational capacity. ReLoRA (49) uses multiple low-rank updates to approximate a final higher-rank update. Other approaches explore elementwise structure for different purposes: FLoRA (**?** ) modulates intermediate activations per task using Hadamard products, and PACE (50) injects multiplicative noise during adaptation. These trends reflect a broader shift toward structured, high-rank updates for improved expressivity.

Our proposed method, ABBA, follows this direction by learning two independent low-rank matrices whose Hadamard product forms the update, enabling high-rank adaptation with full learnability and minimal overhead.

## B    WHY DOES NO CLOSED-FORM RECONSTRUCTION SOLUTION EXIST FOR ABBA?

SVD admits a closed-form solution for low-rank approximation because both the Frobenius and spectral norms are unitarily invariant. This allows the objective:

$$\min_{\text{rank}(X)\leq k} \|M - X\|_F$$

to decouple along singular directions, as guaranteed by the Eckart–Young–Mirsky theorem (16; 17).

In contrast, ABBA solves the problem:

$$\min_{B_1,B_2,A_1,A_2} \|M - (B_1 A_1) \odot (B_2 A_2)\|_F^2, \quad \text{subject to } \text{rank}(B_\ell A_\ell) \leq r,$$

which is a non-convex, quartic optimization problem in the latent factors. The Hadamard product breaks orthogonal invariance; unlike the SVD, the two low-rank matrices $B_1 A_1$ and $B_2 A_2$ cannot be simultaneously diagonalized, and singular directions no longer decouple.

Using Theorem 1, one could, in principle, apply an SVD-like decomposition to $B_1 \odot_r B_2$ and $A_1^\top \odot_r A_2^\top$. However, the rows of these matrices lie on a Segre variety $\mathcal{S}_{r_1,r_2}$[2], meaning they reside in a highly constrained non-linear manifold. In general, such constraints prevent the existence of an exact closed-form solution unless every row lies in a rank-one $\mathbb{R}^{r_1 \times r_2}$ subspace.

As a result, no analogue of truncated SVD exists for this formulation, and optimization must proceed via iterative methods such as gradient descent.

## C    WHY DOES LoRA-STYLE INITIALIZATION OF BOTH ADAPTER PAIRS FAIL?

A naive LoRA-style initialization, where $B_1$ and $B_2$ are initialized to zero while $A_1$ and $A_2$ follow Kaiming uniform initialization, results in training failure due to gradients becoming identically zero. To analyze this, we compute the gradients of the loss with respect to each adapter component and show that they are exactly zero under this initialization, ultimately preventing any learning.

---

[2]The Segre variety $\mathcal{S}_{r_1,r_2} \subset \mathbb{R}^{r_1 r_2}$ is the set of all rank-one tensors expressible as outer products $u \otimes v$, with $u \in \mathbb{R}^{r_1}$, $v \in \mathbb{R}^{r_2}$.

Notice that, for some input, $x$, the output, $y$, is of the form

$$y = Wx + \Delta W x$$
$$= Wx + s_{\text{ABBA}} \left( (B_1 A_1) \odot (B_2 A_2) \right) x$$
$$\mathcal{L} = f(y)$$

To get the final closed form gradients, we compute the gradients element-wise. We can therefore have, for some matrix $Z \in \{A_1, A_2, B_1, B_2\}$,

$$\frac{\partial \mathcal{L}}{\partial Z_{pq}} = \sum_{i,j} \frac{\partial \mathcal{L}}{\partial \Delta W_{ij}} \frac{\partial \Delta W_{ij}}{\partial Z_{pq}}$$

$$= \sum_{i,j} \frac{\partial \mathcal{L}}{\partial \Delta W_{ij}} \frac{\partial \left( s_{\text{ABBA}} \left( (B_1 A_1) \odot (B_2 A_2) \right)_{ij} \right)}{\partial Z_{pq}}$$

$$= s_{\text{ABBA}} \sum_{i,j} G_{ij} \cdot \frac{\partial \left( (B_1 A_1) \odot (B_2 A_2) \right)_{ij}}{\partial Z_{pq}} \tag{9}$$

**For $Z = A_1$.** We need to compute

$$\frac{\partial \mathcal{L}}{\partial A_{1,pq}} = s_{\text{ABBA}} \sum_{i,j} G_{ij} \cdot \frac{\partial \left( (B_1 A_1) \odot (B_2 A_2) \right)_{ij}}{\partial A_{1,pq}}$$

$$= s_{\text{ABBA}} \sum_{i,j} G_{ij} \cdot \frac{\partial \left( (B_1 A_1)_{ij} (B_2 A_2)_{ij} \right)}{\partial A_{1,pq}}$$

$$= s_{\text{ABBA}} \sum_{i,j} G_{ij} (B_2 A_2)_{ij} \frac{\partial \left( (B_1 A_1)_{ij} \right)}{\partial A_{1,pq}}$$

$$= s_{\text{ABBA}} \sum_{i,j} G_{ij} (B_2 A_2)_{ij} \left( \sum_l \frac{\partial \left( B_{1,il} A_{1,lj} \right)}{\partial A_{1,pq}} \right)$$

Notice that $\frac{\partial (B_{1,il} A_{1,lj})}{\partial A_{1,pq}} = 0$ when $l \neq p$ and $j \neq q$ else $= B_{1,ip}$. This means we can rewrite the above summation as

$$\frac{\partial \mathcal{L}}{\partial A_{1,pq}} = s_{\text{ABBA}} \sum_i G_{iq} (B_2 A_2)_{iq} B_{1,ip}$$

$$= s_{\text{ABBA}} \sum_i (G \odot (B_2 A_2))_{iq} B_{1,ip}$$

Notice that the above equation is nothing but a inner product of the $p^{\text{th}}$ row of $B_1^\top$ and $q^{\text{th}}$ column of $(G \odot (B_2 A_2))$. We can therefore write the following

$$\frac{\partial \mathcal{L}}{\partial A_1} = s_{\text{ABBA}} B_1^\top \left( G \odot (B_2 A_2) \right) \tag{10}$$

**For $Z = B_1$.** Following the same analysis as we did for $A_1$, we have the following for $B_1$.

$$\frac{\partial \mathcal{L}}{\partial B_{1,pq}} = s_{\text{ABBA}} \sum_{i,j} G_{ij} (B_2 A_2)_{ij} \left( \sum_l \frac{\partial \left( B_{1,il} A_{1,lj} \right)}{\partial B_{1,pq}} \right)$$

Again notice that $\frac{\partial (B_{1,il} A_{1,lj})}{\partial B_{1,pq}} = 0$ when $i \neq p$ and $l \neq q$ else $= A_{1,qj}$. This means we can rewrite the above summation as

$$\frac{\partial \mathcal{L}}{\partial B_{1,pq}} = s_{\text{ABBA}} \sum_j G_{pj} (B_2 A_2)_{pj} A_{1,qj}$$

$$= s_{\text{ABBA}} \sum_j (G \odot (B_2 A_2))_{pj} A_{1,qj}$$

Notice that the above equation is nothing but a inner product of the $p^{\text{th}}$ row of $(G \odot (B_2 A_2))$ and $q^{\text{th}}$ column of $A_1^\top$. We can therefore write the following

$$\frac{\partial \mathcal{L}}{\partial B_1} = s_{\text{ABBA}}(G \odot (B_2 A_2))A_1^\top \tag{11}$$

It is also easy to see that our formulation is symmetric in $A_i$'s and $B_i$'s implying we also have

**For $Z = B_2$.** Following Eqn. (11)

$$\frac{\partial \mathcal{L}}{\partial B_2} = s_{\text{ABBA}}(G \odot (B_1 A_1))A_2^\top \tag{12}$$

**For $Z = A_2$.** Following Eqn. (10),

$$\frac{\partial \mathcal{L}}{\partial A_2} = s_{\text{ABBA}} B_2^\top (G \odot (B_1 A_1)) \tag{13}$$

It is clear that initializing both $B_1$ and $B_2$ to zero causes all gradients to become zero, thereby preventing any learning and leading to complete training failure.

In this section, we provide the proofs for the assertions from the main text.

## D    PROOF OF THEOREM 2: RANK-STABILITY OF ABBA

**Theorem** (Rank-Stability of ABBA). *Let the ABBA update be $\Delta W = s_{ABBA} (B_1 A_1) \odot (B_2 A_2)$, with $B_1, A_1, B_2, A_2$ independent, mean-zero random matrices with finite variance. Then, the update satisfies the stability conditions of Definition 1 if and only if:*

$$s_{ABBA} \in \Theta\left(\frac{1}{\sqrt{r_1 r_2}}\right).$$

**Forward $2^{\text{nd}}$ moment (ABBA adapter).**    For any input $x \in \mathbb{R}^{d_{\text{in}}}$, the ABBA adapter produces

$$y_i = s_{\text{ABBA}} \sum_{j=1}^{d_{\text{in}}} M_{1,ij} M_{2,ij} \cdot x_j, \quad \forall i \in \{1, \ldots, d_{\text{out}}\}, \tag{14}$$

where $M_1 = B_1 A_1 \in \mathbb{R}^{d_{\text{out}} \times d_{\text{in}}}$ and $M_2 = B_2 A_2 \in \mathbb{R}^{d_{\text{out}} \times d_{\text{in}}}$. Thus

$$\mathbb{E}[y_i^2] = s_{\text{ABBA}}^2 \cdot \mathbb{E}\left[\left(\sum_{j=1}^{d_{\text{in}}} M_{1,ij} M_{2,ij} x_j\right)^2\right].$$

**Gradient computation.**    The gradient with respect to the input is

$$(\nabla_x L)_j = \sum_{i=1}^{d_{\text{out}}} \frac{\partial L}{\partial y_i} \frac{\partial y_i}{\partial x_j} = s_{\text{ABBA}} \sum_{i=1}^{d_{\text{out}}} M_{1,ij} M_{2,ij} (\nabla_y L)_i,$$

where $(\nabla_y L)_i = \frac{\partial L}{\partial y_i}$. Hence

$$\mathbb{E}[(\nabla_x L)_j^2] = s_{\text{ABBA}}^2 \cdot \mathbb{E}\left[\left(\sum_{i=1}^{d_{\text{out}}} M_{1,ij} M_{2,ij} (\nabla_y L)_i\right)^2\right].$$

**Second-moment factorization.**    Since $M_1$ and $M_2$ are independent with mean-zero entries,

$$\mathbb{E}[M_{1,ij}^2] = r_1 \, \mathbb{E}[B_{1,11}^2]\mathbb{E}[A_{1,11}^2], \quad \mathbb{E}[M_{2,ij}^2] = r_2 \, \mathbb{E}[B_{2,11}^2]\mathbb{E}[A_{2,11}^2].$$

Therefore

$$\mathbb{E}[M_{1,ij}^2 M_{2,ij}^2] = \mathbb{E}[M_{1,ij}^2] \, \mathbb{E}[M_{2,ij}^2] = r_1 r_2 \cdot \mathbb{E}[B_{1,11}^2]\mathbb{E}[A_{1,11}^2]\mathbb{E}[B_{2,11}^2]\mathbb{E}[A_{2,11}^2].$$

**Forward and backward moments.** Substituting,

$$\mathbb{E}[y_i^2] = s_{\text{ABBA}}^2 \cdot d_{\text{in}} \cdot r_1 r_2 \cdot C \cdot \mathbb{E}[x_j^2],$$
$$\mathbb{E}[(\nabla_x L)_j^2] = s_{\text{ABBA}}^2 \cdot d_{\text{out}} \cdot r_1 r_2 \cdot C \cdot \mathbb{E}[(\nabla_y L)_i^2],$$

where $C = \mathbb{E}[B_{1,11}^2]\mathbb{E}[A_{1,11}^2]\mathbb{E}[B_{2,11}^2]\mathbb{E}[A_{2,11}^2]$.

**Rank-stability.** By Definition 1, forward stability requires $\mathbb{E}[y_i^2] = \Theta_{r_1,r_2}(1)$ whenever $\mathbb{E}[x_j^2] = \Theta(1)$, and backward stability requires $\mathbb{E}[(\nabla_x L)_j^2] = \Theta_{r_1,r_2}(1)$ whenever $\mathbb{E}[(\nabla_y L)_i^2] = \Theta(1)$. Both hold exactly when

$$s_{\text{ABBA}}^2 r_1 r_2 = \Theta(1),$$

that is,

$$s_{\text{ABBA}} \in \Theta\left(\frac{1}{\sqrt{r_1 r_2}}\right).$$

Following Definition 1, these results imply that ABBA adapters are rank-stabilized.

# E SELECTING $\alpha$

We perform a sweep over different $\alpha$ values, reporting results for Mistral-7B in Table 8. In line with prior results in Table 4, ABBA performs best when $\alpha_{\text{LoRA}}$ lies in the typical range of 16–32.

Table 8: Performance comparison across different $\alpha$ values for Mistral-7B.

| $\alpha$ | Accuracy (↑) | |
|---|---|---|
| | GSM8K | MATH |
| 4 | 62.17 | 17.10 |
| 8 | 64.06 | 17.60 |
| 12 | 64.43 | 18.14 |
| 16 | 66.10 | 18.16 |
| 24 | 66.26 | 18.08 |
| 32 | 65.81 | 17.68 |
| 48 | 65.20 | 16.82 |
| 64 | 64.79 | 15.08 |

# F EFFECT OF VARYING RANK

We evaluate ABBA on Mistral-7B while varying the total rank budget $r$, with $r_1 = r_2 = r/2$ and results shown in Table 9. Performance improves substantially from $r = 16$ to $r = 32$, with the best results achieved at $r = 32$, after which gains begin to saturate. This aligns with our hypothesis: at $r = 32$, ABBA's effective rank reaches $r_1 \times r_2 = 16 \times 16 = 256$, which is significantly higher than the effective rank of 64 at $r = 16$. The increased expressivity enables ABBA to better capture task-specific patterns. However, increasing the rank beyond 32, or an effective rank beyond 256, yields diminishing returns and may lead to overfitting, mirroring trends observed in using very high-ranked LoRA as well.

Table 9: Performance comparison of ABBA on Mistral-7B across varying total rank values $r$.

| Rank | Accuracy (↑) | |
|---|---|---|
| | GSM8K | MATH |
| 16 | 64.97 | 18.06 |
| 32 | **66.26** | **18.08** |
| 64 | 65.05 | 17.98 |
| 128 | 65.73 | 17.96 |

## G  Training Time

Following the discussion in Section 4.4, we report training times across all models and tasks in Table 10. ABBA incurs only a negligible overhead of approximately $2 - 3\%$ compared to LoRA, primarily due to the extra computation resulting from the Hadamard product.

Table 10: Training time comparison between ABBA and LoRA across multiple model and task settings.

| Model | Training Time | |
|---|---|---|
| | LoRA | ABBA |
| Llama-3.2 1B (Commonsense) | 2:42:17 | 2:46:18 |
| Llama-3.2 3B (Commonsense) | 6:05:43 | 6:11:26 |
| Mistral-7B (Arithmetic) | 1:15:55 | 1:18:22 |
| Gemma-2 9B (Arithmetic) | 1:45:33 | 1:49:45 |

## H  Performance on Coding Tasks

We also evaluate ABBA against other PEFT variants on code generation tasks. We fine-tuned Llama-3.2-1B on a 100K subset of the CodeFeedback dataset (51) and assessed performance using Pass@1 on HumanEval (52). As shown in Table 11, ABBA again performs very strongly, outperforming all other PEFT variants and even full fine-tuning (all PEFT methods are run at rank 32). These results demonstrate that the advantages of our method generalize to more complex tasks such as code generation.

Table 11: Comparison of multiple FT methods across the coding benchmark - HumanEval. Best results among PEFT methods are in **bold**.

| Method | # Params | HumanEval ($\uparrow$) |
|---|---|---|
| Full FT | 1.24 B | 23.17 |
| LoRA | 22.54 M | 20.73 |
| HiRA | 22.54 M | 21.95 |
| ABBA | 22.54 M | **25.61** |

## I  Rank-Stability of ABBA: Gradient Norms

In Theorem 2, we had established the optimal scaling for ABBA,

$$s_{\text{ABBA}} \in \Theta\left(\frac{1}{\sqrt{r_1 r_2}}\right).$$

We now verify this result empirically by comparing several scaling strategies:

$$s_{\text{ABBA}} \in \Theta(1), \quad \Theta\left(\frac{1}{(r_1 r_2)^{1/6}}\right), \quad \Theta\left(\frac{1}{(r_1 r_2)^{1/4}}\right), \quad \Theta\left(\frac{1}{(r_1 r_2)^{1/2}}\right).$$

As shown in Figure 5, the scaling $s_{\text{ABBA}} \in \Theta\left(\frac{1}{\sqrt{r_1 r_2}}\right)$ produces the most stable gradient norms, consistent with our theoretical findings.

## J  Stability in Chained ABBA: Gradient Norms

We present gradient norms for the chained ABBA setup (with four adapter pairs) compared to the conventional ABBA configuration (with two pairs) in Figure 6. As hypothesized in Section 4, the chained variant exhibits significantly greater gradient-norm instability than the standard ABBA model.

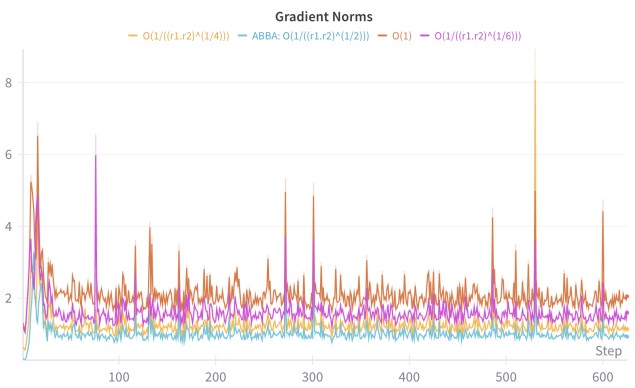

Figure 5: Gradient norms in Mistral-7B comparing various scaling strategies.

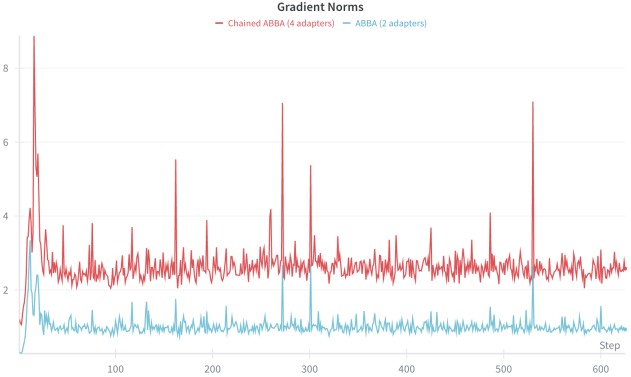

Figure 6: Gradient norms in Mistral-7B comparing chained ABBA (4 pairs) and normal ABBA

## K    COMPARISON WITH HIGH-RANK BASELINES

We evaluate ABBA against additional higher-rank PEFT variants, namely Flora (53), MoRA (**?** ), and KronA (48), on code generation tasks. We fine-tuned Llama-3.1 8B on a 100K subset of the CodeFeedback dataset (51) and assessed performance using Pass@1 on HumanEval (52). As shown in Table 12, ABBA again performs very strongly, outperforming all other higher-rank PEFT variants.

Table 12: Comparison of multiple PEFT methods across the coding benchmark - HumanEval. Best results among PEFT methods are in **bold**.

| Method | HumanEval (↑) |
|--------|---------------|
| Full FT | 48.54 |
| LoRA | 45.73 |
| KronA | 45.12 |
| Flora | 45.56 |
| MoRA | 47.28 |
| HiRA | 48.17 |
| ABBA | **49.39** |

## L    EXPERIMENTAL DETAILS

We implement all models using PyTorch (54) and HuggingFace Transformers (55). All experiments run on a single NVIDIA A6000 GPU (48 GB). To reduce memory usage, we initialize base models in `torch.bfloat16` precision. We train each configuration with the AdamW optimizer (56) and report the mean performance over three random seeds.

We configure Llama-3.2 1B, Llama-3.2 3B, Mistral-7B, and Gemma-2 9B using the hyperparameters shown in Table 13. We conduct a sweep over learning rates and scaling factors to identify optimal settings for each model-task pair. ABBA generally performs better with slightly higher learning rates compared to LoRA, and we recommend initiating hyperparameter sweeps in that range.

While we adopt most settings from prior work (27), we perform a targeted learning rate sweep to optimize performance. For baseline comparisons, we replicate the experimental setups from the original PiSSA (26), rsLoRA (21), DoRA (11), LoRA-Pro (14), and HiRA (13) papers to ensure fair and consistent evaluation.

Table 13: Hyperparameter settings for training Llama-3.2 1B and 3B on COMMONSENSE170K, and Mistral-7B and Gemma-2 9B on MetaMathQA.

|  | Llama-3.2 1B / 3B | Mistral-7B / Gemma-2 9B |
|--|-------------------|-------------------------|
| Optimizer | AdamW | AdamW |
| Batch size | 6 | 1 |
| Max. Seq. Len | 256 | 512 |
| Grad Acc. Steps | 24 | 32 |
| Epochs | 2 | 1 |
| Dropout | 0.05 | 0 |
| Learning Rate | $1 \times 10^{-3}$ | $1 \times 10^{-3}$ |
| LR Scheduler | Linear | Cosine |
| Warmup Ratio | 0.02 | 0.02 |

## M    DATASET DETAILS

**COMMONSENSE170K** is a unified benchmark that aggregates eight commonsense reasoning datasets into a single multi-task setting (27). Each instance is a multiple-choice question, and models are prompted to select the correct answer without providing explanations. We adopt the prompt format introduced by the paper (27). Below, we briefly describe the constituent datasets:

- **OBQA** (28): Open-book QA requiring retrieval and multi-hop reasoning over external knowledge.
- **ARC Challenge (ARC-c)** (29): Difficult grade-school science questions designed to test advanced reasoning beyond surface heuristics.
- **ARC Easy (ARC-e)** (29): Simpler science questions assessing core factual and conceptual understanding.
- **WinoGrande** (30): Pronoun resolution tasks requiring commonsense inference to resolve ambiguity.
- **HellaSwag** (31): Next-sentence prediction under a constrained completion setting, testing grounded understanding of everyday scenarios.
- **PIQA** (32): Physical reasoning tasks where models select the most sensible solution to a practical problem.
- **SIQA** (33): Social reasoning benchmark involving questions about intent, social dynamics, and consequences of human actions.
- **BoolQ** (34): Binary (yes/no) questions drawn from natural queries, requiring contextual understanding of short passages.

**MetaMathQA** (35) reformulates existing mathematical problems into alternative phrasings that preserve their original semantics, offering diverse surface forms without introducing new information. We evaluate models fine-tuned on this dataset using two benchmarks: **GSM8K** (36), which targets step-by-step reasoning in elementary arithmetic word problems, and **MATH** (37), which features high-difficulty problems drawn from math competitions. We evaluate solely based on the correctness of the final numeric answer.

# N    USE OF LARGE LANGUAGE MODELS

LLMs are used exclusively for minor writing assistance, for example, to enhance grammar and improve sentence clarity.

