# OpenReview forum: "ABBA-Adapters: Efficient and Expressive Fine-Tuning of Foundation Models"
_ICLR.cc/2026/Conference — ICLR 2026 Poster_

### Official Review · Reviewer_Z4En · 2025-10-21

**Soundness:** 3
**Presentation:** 3
**Contribution:** 2
**Rating:** 4
**Confidence:** 4

**Summary:**

This paper proposed ABBA, a variant of LoRA that relies on the Hadamard product of two learnable low-rank matrices. The key difference compared to HiRA is that both terms in the Hadamard product are learnable, and are decomposed through LoRA. In doing so, the Hadamard product can be calculated through the well-known Khatri–Rao factorization (or equivalently, face-splitting product). Experiments are conducted on commonsense reasoning datasets with LLaMA3.2 1B and 3B, and Arithmetic reasoning with Mistral-7B and Gemma-2 9B, demonstrating the empirical advantages of the advocated approach.

**Strengths:**

1. The writing is easy to follow and the presentation is clear
2. The Khatri–Rao factorization provides a computationally efficient approach to bypass the Hadamard product.
3. The rank stability is confirmed theoretically.
4. Numerical results are promising, which demonstrate a consistent performance boot.

**Weaknesses:**

1. My major concern is the computational cost of ABBA. With the face-splitting (i.e., row-wise Khatri-Rao) product in Theorem 1, the forward and backpropagation cost of $\Delta W x$ is $\mathcal{O}((m+n)r_1r_2) = \mathcal{O}((m+n)r^2)$, which is $\mathcal{O}(r)$ times higher than vanilla LoRA. In other words, ABBA is be less scalable with the rank increasing. It would be beneficial if experiments can be conducted to showcase how the time and space complexities change with r (e.g., from $r=4$ to $r=512$) under different model sizes.
2. There is no need to provide a proof for Theorem 1 in Appendix D.1, as it is a well-known result in linear algebra, and the citation (18) could be sufficient.
3. The "ABBA vs. LoRA: Reconstruction" paragraph in Section 2.4 is not convincing enough, given the lack of tight theoretical bounds. Could you please provide some insights regarding how the bound can be improved?
4. The models used in Section 4 are relatively small (<10B). It would be beneficial if experiments can be scaled to larger models such as Gemma-3-27B, and more complicated tasks such as coding.

**Questions:**

See weaknesses above.

---

> ### Author Response · Authors · 2025-11-20
>
> Thank you for your detailed review. We address each concern below and present **additional experimental results** to strengthen the claims in our paper.
>
> `W1: “My major concern is the computational cost of ABBA. With the face-splitting (Khatri-Rao) product in Theorem 1, the forward and backpropagation cost of Delta W x is O((m+n)r1.r2) = O((m+n)r^2), which is O(r) times higher than vanilla LoRA. In other words, ABBA is less scalable with the rank increasing. It would be beneficial if experiments can be conducted to showcase how the time and space complexities change with r under different model sizes.”`
>
>
> We agree that, if one considers the adapter in isolation, the forward/backward cost of $(\Delta W x)$ under the row-wise Khatri–Rao factorization is $\mathcal{O}((m+n), r_1 r_2),$ which is $\mathcal{O}(r)$ larger than the $\mathcal{O}((m+n), r)$ cost of a vanilla LoRA adapter when we set $r_1 = r_2 = r/2$.
>
> However, this $\mathcal{O}((m+n), r_1 r_2)$ term describes only the extra cost of the low-rank update $(\Delta W).$ In a full transformer layer the dominant cost remains the base projection $(W x)$ with forward+backward cost $\mathcal{O}(mn)$ together with attention and MLP blocks. Both LoRA and ABBA incur this $\mathcal{O}(mn)$
> backbone cost; they differ only in the comparatively smaller adapter term. As a result, the overall per-layer cost of ABBA is only modestly larger than that of LoRA in the practical rank regime we use.
>
> ----
>
> To make this concrete, we added experiments that directly measure both space and time complexity as a function of adapter rank across several model sizes, comparing full fine-tuning, LoRA, and ABBA:
>
> **Space (GPU memory)**
>
> We find that the peak memory usage of ABBA and LoRA remains nearly identical up to r=128. These results are shown in **Figure 9 (Appendix L)** of the updated paper.
>
> **Timing (latency / throughput)**
>
> We similarly observe that training-time characteristics (latency per step and tokens/sec) of ABBA and LoRA remain very close up to r=128. These results are shown in **Figure 7 and 8 (Appendix K)** in the updated manuscript.
>
> ---
>
> We want to emphasize that a key benefit of ABBA is that it achieves a much larger effective rank than LoRA for the same r.
> For example, at r=128, our method realizes an effective rank of 4096, whereas LoRA has an effective rank of 128. This shows that ABBA offers substantially higher expressive capacity per unit of memory for only a modest increase in compute.
>
> Finally, our method is not intended to be used at extremely large adapter ranks (e.g., r=512). Since ABBA already attains very high effective rank at moderate adapter ranks (e.g., $r \leq 128$), there is no need to scale r further in practice. In this range, our computational efficiency, both in terms of space and time, is essentially the same as LoRA, while achieving performance that surpasses other PEFT methods and even full fine-tuning.
>
>
>
> `W2: “There is no need to provide a proof for Theorem 1 in Appendix D.1, as it is a well-known result in linear algebra, and the citation (18) could be sufficient.”`
>
> Thank you for the suggestion. We had originally included the proof because the Khatri–Rao formulation is important for explaining the memory-efficient implementation of our method.
>
> However, following your recommendation, we have removed the proof and now simply include the citation.

---

> > ### Author Response · Authors · 2025-11-20
> >
> > `W3: “The "ABBA vs. LoRA: Reconstruction" paragraph in Section 2.4 is not convincing enough, given the lack of tight theoretical bounds. Could you please provide some insights regarding how the bound can be improved?”`
> >
> > Thank you for raising this important point. We clarify the theoretical landscape around reconstruction bounds for ABBA. Unlike the classical low-rank approximation problem
> > $$
> > \min_{\text{rank}(X)\le r} \|M - X\|_F^2,
> > $$
> > where the Eckart–Young theorem gives a closed-form SVD optimum, the ABBA decomposition
> > $$M \approx (B_1A_1)\odot(B_2A_2)$$  yields a quartic, non-convex optimization problem.
> >
> > Prior work (Ciaperoni et al., 2024; Wertz et al., 2025) establishes that **no closed-form optimal solution is known** for Hadamard-structured factorizations, and all practical methods rely on numerical optimization. While this prevents deriving SVD-tight optimality results, we can still use classical singular-value reasoning to obtain meaningful bounds.
> >
> > Specifically, ABBA can always represent any rank-$r$ matrix trivially (by setting $B_2A_2=\mathbf{1}$), and in the best case it can represent matrices of rank up to $r^2$. Thus the ABBA reconstruction error must lie between the optimal SVD errors of rank-$r$ and rank-$r^2$ approximations:
> > $$
> > \sum_{i=r+1}^{\min(m,n)} \sigma_{i}^2 \geq\ \min_{ABBA} \|M - \Delta W\|_F^2
> > $$
> >
> > and
> >
> > $$
> > \sum_{i=r^2+1}^{\min(m,n)} \sigma_{i}^2 \le\ \min_{ABBA} \|M - \Delta W\|_F^2
> > $$
> >
> > where $\sigma_i$ are the singular values of $M$. These bounds show that the achievable error depends on the singular value decay of the target matrix, but they are inherently loose because the Hadamard factorization problem has no SVD-style optimal solution. This looseness is exactly why recent work (Ciaperoni et al., 2024; Wertz et al., 2025) uses **empirical reconstruction** as the standard methodology and finds that Hadamard/Khatri–Rao factorizations consistently achieve reconstruction errors **far below** what worst-case bounds predict.
> >
> > Our results follow this established approach: empirical reconstruction across multiple matrix classes shows that ABBA reliably outperforms LoRA under identical parameter budgets, even though tight closed-form bounds are provably out of reach for this class of decompositions. We hope this clarifies why the theoretical bounds cannot be strengthened in a closed-form manner and why empirical reconstruction remains the accepted and informative tool for evaluating expressivity in this setting.
> >
> >
> >
> > `W4: “The models used in Section 4 are relatively small (<10B). It would be beneficial if experiments can be scaled to larger models such as Gemma-3-27B, and more complicated tasks such as coding.”`
> >
> > Thanks for the suggestion.
> >
> > We now evaluate ABBA in comparison with other PEFT methods on code generation. Specifically, we fine-tuned Llama-3.2-1B on a 100K subset of the CodeFeedback dataset and evaluated performance using Pass@1 on HumanEval. As shown in the results below, ABBA again performs very strongly, outperforming all other PEFT variants and even full fine-tuning (all PEFT methods are run at rank 32). This indicates that the benefits of our method generalize to more challenging tasks such as code generation.
> >
> > We include these results in **Table 10 (Appendix H)** of the updated paper.
> >
> >
> > $$
> > \\begin{array}{|l|c|c|}
> > \\hline
> > \\textbf{Method} &
> > \\begin{array}{c}\\textbf{\\# Params}\\end{array} &
> > \\begin{array}{c}\\textbf{HumanEval}\\;(\uparrow)\\end{array} \\\\
> > \\hline
> > \\text{Full FT}   & 1.24\\text{B}  & 23.17 \\\\
> > \\text{LoRA}      & 22.54\\text{M} & 20.73 \\\\
> > \\text{HiRA}      & 22.54\\text{M} & 21.95 \\\\
> > \\text{ABBA} & 22.54\text{M} & \mathbf{25.61} \\\\
> > \\hline
> > \\end{array}
> > $$
> >
> >
> >
> > ---
> >
> > Due to hardware constraints (a single 48GB GPU), we are unable to run fine-tuning experiments on models larger than 10B parameters, as they do not fit in memory. We therefore cannot provide results for Gemma-3-27B or similarly sized models, and we sincerely apologize for this.
> >
> > We note, however, that our experimental scale aligns with prior PEFT work, including HiRA, MoRA, rsLoRA, LoRA-GA, and LoRA-Pro, which evaluate models up to the 7B–9B range. We believe that our analysis at this commonly adopted scale is informative and representative of the method’s behavior.
> >
> > ---
> >
> > ### **References**
> >
> > - Ciaperoni, M., Gionis, A., & Mannila, H. (2024). *The Hadamard decomposition problem*. Data Mining and Knowledge Discovery, 38(4), 2306–2347.
> > - Wertz, S., Vandaele, A., & Gillis, N. (2025). *Efficient algorithms for the Hadamard decomposition*. arXiv:2504.13633.
> >
> > ---
> >
> > Thank you again for the helpful suggestions that have helped improve our claims. If our rebuttal addresses the issues raised, we kindly ask that you **consider raising your score**. We are happy to clarify further questions you might have!

---

> > > ### Author Response · Authors · 2025-11-26
> > >
> > > This is a gentle reminder regarding our rebuttal. We understand how busy the reviewing schedule can be and apologize if this was already on your list to address. Your thoughtful engagement with our work is sincerely appreciated, and we hope our responses covered your concerns as fully as possible. Thank you again for the detailed review that helped us strengthen our claims. If anything else comes up, we would be happy to clarify!

---

> > > > ### Comment · Reviewer_Z4En · 2025-11-26
> > > >
> > > > Thank you for the detailed clarification. After carefully evaluating the authors’ rebuttal, I have decided to maintain my preliminary score of 4 for now. My main reservation remains the novelty of ABBA relative to HiRA. In addition, the claimed empirical advantages of ABBA are not yet convincingly demonstrated on more complex tasks with larger models. For example, the additional experiments provided in the rebuttal is still restricted to a small model with merely 1B parameters. The evaluation also lacks a clear apple-to-apple comparison against other high-rank baselines such as FLoRA [1], ReLoRA [2], MoRA [3], and ScaLoRA [4], despite including several less relevant baselines (e.g., rsLoRA, PiSSA, DoRA, LoRA-Pro).
> > > >
> > > > That said, I now consider the paper **borderline (around a score of 5)**, and **I am open to raising my score as I am leaning toward acceptance**.
> > > >
> > > > [1] Flora: Low-rank adapters are secretly gradient compressors, ICML’24.
> > > > [2] ReLoRA: High-Rank Training Through Low-Rank Updates, ICLR’24.
> > > > [3] Mora: High-rank updating for parameter-efficient fine-tuning, arXiv preprint’24.
> > > > [4] ScaLoRA: Optimally Scaled Low-Rank Adaptation for Efficient High-Rank Fine-Tuning, arXiv preprint’25.

---

> ### Author Response · Authors · 2025-11-27
>
> We thank the reviewer for the detailed feedback and for being open to revising their score. We address each of the points raised below.
>
>
> `Clarifying novelty of ABBA relative to HiRA`
>
> While ABBA and HiRA both introduce multiplicative structure, the key contribution of ABBA is a *different parameterization principle*. HiRA modulates the pretrained weight $W_0$ elementwise by a learned low-rank matrix, which makes all updates inherently tied to the structure of $W_0$. This coupling is a strong structural constraint: if the target update is not well-aligned with $W_0$, HiRA’s representable space becomes significantly restricted.
>
> ABBA removes this constraint entirely by learning two independent low-rank matrices whose Hadamard interaction defines the update:
> $$
> \Delta W = (B_1A_1)\odot(B_2A_2).
> $$
>
> This decoupling yields a representational geometry that HiRA cannot reach, while still retaining the benefits of multiplicative updates. The non-trivial aspect is that such multiplicative low-rank interactions would normally induce a high-rank intermediate object; ABBA becomes viable only because the Khatri–Rao structure allows us to express this interaction with LoRA-like memory and compute. This leads to a function class strictly larger than LoRA or HiRA under the same parameter budget, and crucially, a different optimization landscape, since gradients propagate through two separate low-rank factors rather than a single modulation of $W_0$.
>
> For these reasons, ABBA is not simply a variation on HiRA: it breaks the dependency on $W_0$, enables higher effective rank without increasing parameters, and introduces an optimization geometry that HiRA does not access. We hope this clarifies why we view ABBA as a distinct and meaningful contribution.
>
> `Comparison with additional baselines on larger models with complex tasks`
>
>
> In response to your requests, we now include experiments with a bigger model on complex tasks, ie. **Llama 3.1 8B on the code generation task** described previously. This is the largest model size we could use given our constraints, but it is still consistent with the PEFT literature, and we do not expect this to affect the validity of our results.
>
> We have added new comparisons of ABBA with high rank baselines, namely **Flora**, **MoRA**, and **KronA**, in addition to HiRA (as also asked by reviewer XkLT). Due to time constraints, we are not able to run experiments on all papers mentioned by the reviewer. We also note that ReLoRA requires full fine tuning for 25 percent of the training time, which leads to memory issues and is more commonly used for efficient pretraining. ScaLoRA was released on Arxiv on 27 Oct 2025, after the ICLR deadline, so we are not able to include comparisons with it at this time, again due to time limitations.
>
> We present results comparing ABBA with LoRA, HiRA, Flora, MoRA, and KronA in the table below. These results show a consistent and very promising trend, with **ABBA continuing to outperform the additional high rank baselines**, along with the baselines already evaluated in the paper.
>
> $$
> \\begin{array}{|l|c|c|}
> \\hline
> \\textbf{Method} &
> \\begin{array}{c}\\textbf{HumanEval}\\;(\uparrow)\\end{array} \\\\
> \\hline
> \\text{Full FT}   & 48.54 \\\\
> \\text{LoRA}  & 45.73 \\\\
> \\text{KronA}  & 45.12 \\\\
> \\text{Flora} & 45.56 \\\\
> \\text{MoRA}   & 47.28 \\\\
> \\text{HiRA}    & 48.17 \\\\
> \\text{ABBA} & \mathbf{49.39} \\\\
> \\hline
> \\end{array}
> $$
>
> ---
>
>
> We believe that these comparisons further support ABBA’s position as a strong high-rank PEFT method. We kindly request that you consider increasing your score if your major concerns have been addressed. Thank you again for your time.

---

> > ### Comment · Reviewer_Z4En · 2025-11-27
> >
> > Thank you for the follow-up elaboration. My concerns have been adequately addressed. I'm raising my score to 8.

---

> > > ### Author Response · Authors · 2025-11-27
> > >
> > > Thank you again for the scientific discussions. We are glad that our rebuttal was able to adequately address your concerns, and we appreciate you raising your score.

---

> > > > ### Author Response · Authors · 2025-12-02
> > > >
> > > > Thank you again for your review and discussions! We have included comparisons with the additional high-rank baselines in **Table 12 (Appendix M)**.

---

### Official Review · Reviewer_FALt · 2025-10-27

**Soundness:** 3
**Presentation:** 4
**Contribution:** 3
**Rating:** 6
**Confidence:** 4

**Summary:**

- The paper proposes an alternative PEFT method to LoRA called ABBA
- ABBA paramtrizes the residual weights $\Delta W$ used during the fine-tuning as $\Delta W = s(B_1 A_1) \odot (B_2 A_2)$ where $A_1, A_2, B_1, B_2$ are low rank matrices of ranks $r_1$ and $r_2$ respectively, where $s=\frac{\alpha^2}{r_1r_2}$ for rank stabilization.
- $A_1$ and $B_1$ are initialized using the truncated SVD of the pretrained weights $W_o$ by keeping the top $r_1$ singular values and vectors.
- $A_2$ and $B_2$ are initialized using the the standatd LoRA initialization scheme.
- ABBA uses Khatri-Rao factorization to rewrite the paramtrization as $(B_1A_1) \odot (B_2A_2) = (B_1 \odot_r B_2)(A_1 \odot_r A_2)$ where $\odot_r$ is the Khatri-Rao product, makig the computation efficient. Here, $B_1 \odot_r B_2 = B_{kr} \in \mathbb{R}^{m \times r_1 r_2}$ and $A_1 \odot_r A_2 = A_{kr} \in \mathbb{R}^{r_1 r_2 \times n}$.

**Strengths:**

1. The method a clear theoretical motivation.
2. The Khatri-Rao formulation makes the method computationally efficient, and hence practically feasible.
3. The method is easy to implement and can be easily integrated with existing PEFT methods.
4. The method is evaluated on commonsense reasoning, arithmetic reasoning and outperforms prior PEFT methods across model sizes.
5. The method is ablated well to study its properties (initialization strategies, scaling factors, layer placement, and chaining more low-rank matrices instead of only two).

**Weaknesses:**

1. The paper compares ABBA with total ranks of 16 and 32 with rank 32 LoRA (and variants). However, LoRA can have onptimization problems with larger ranks, and many times, using smaller ranks can lead to slightly better performance. Hence, comparison with LoRA should also be done by setting the LoRA rank to 16 for a more thorough comparison.
2. I find it hard to believe that full fine-tuning lags behind ABBA by such a large margin. While the authors try to justify this on lines 330-332, better hyperparameter tuning (especially setting a low learning rate) can possibly help full fine-tuning perform better. This does not necessarily mean that ABBA does not have merits over full fine-tuning, but rather that the expectations should be tempered.
3. Continuing the point above, the experiments on arithmetic reasoning have been performed with a 20k subset of MetaMathQA. It's possible that with increasing the dataset size, full fine-tuning and LoRA can perform better. Again, this experiment would provide a more comprehensive comparison and help understand the merits of ABBA in limited data settings (wich are often observed in real-world applications).

**Questions:**

1. What is the hyperparameter tuning strategy used for full fine-tuning?
2. Can the authors perfrom the experiments on arithmetic reasoning with a larger dataset, e.g., 40k subset of MetaMathQA?

---

> ### Author Response · Authors · 2025-11-20
>
> We thank the reviewer for the positive assessment and constructive suggestions. Below, we respond to each point raised and provide **additional experiments** to further support our conclusions.
>
> `W1: “The paper compares ABBA with total ranks of 16 and 32 with rank 32 LoRA (and variants). However, LoRA can have optimization problems with larger ranks, and many times, using smaller ranks can lead to slightly better performance. Hence, comparison with LoRA should also be done by setting the LoRA rank to 16 for a more thorough comparison.”`
>
>
> Thank you for raising this point. In addition to the LoRA (r=32) results already reported in the paper, we also evaluate LoRA (r=16) on the same models and datasets.
>
> $$
> \\begin{array}{|l|cc|cc|}
> \\hline
> \\textbf{Method} &
> \\begin{array}{c}\\textbf{Mistral-7B}\\\\\\textbf{GSM8K}\\end{array} &
> \\begin{array}{c}\\textbf{Mistral-7B}\\\\\\textbf{MATH}\\end{array} &
> \\begin{array}{c}\\textbf{Gemma-2\\,9B}\\\\\\textbf{GSM8K}\\end{array} &
> \\begin{array}{c}\\textbf{Gemma-2\\,9B}\\\\\\textbf{MATH}\\end{array}\\\\
> \\hline
> \\text{LoRA (r=16)}   & 61.43 & 15.63 & 75.67 & 36.21\\\\
> \\text{LoRA (r=32)}  & 61.94 & 15.98 & 76.19 & 36.56\\\\
> \\text{ABBA (r=16)} & 64.97 & 18.06 & 78.70 & 38.41\\\\
> \\text{ABBA (r=32)} & \\mathbf{66.26} & \\mathbf{18.08} & \\mathbf{79.76} & \\mathbf{39.18}\\\\
> \\hline
> \\end{array}
> $$
>
> Across all settings, LoRA (r=16) performs slightly worse than LoRA (r=32), and remains significantly below ABBA, even under identical parameter budgets (e.g., ABBA (r=16) vs. LoRA (r=16)). This behavior is consistent with prior findings that LoRA’s performance on reasoning-heavy tasks tends to improve with moderate rank values.
>
> Thus, this shows that using LoRA (r=32) in our main comparisons does not disadvantage LoRA, and ABBA’s performance gains are not attributable to LoRA being evaluated at a suboptimal rank.
>
>
>
>
> `W2 & Q1: Clarifications around full fine-tuning results. W2: “I find it hard to believe that full fine-tuning .. This does not necessarily mean that ABBA does not have merits over full fine-tuning, but rather that the expectations should be tempered.” Q1: “What is the hyperparameter tuning strategy used for full fine-tuning?”`
>
> We agree that full fine-tuning is sensitive to hyperparameters, and we acknowledge that our full-FT configuration was not extensively tuned. Since full FT is substantially more expensive than PEFT, we relied on standard, publicly recommended configurations from prior work such as LoRA, DoRA, HiRA, and PiSSA. These are generally regarded as strong defaults for small-to-medium arithmetic and commonsense reasoning datasets, but they are not necessarily optimal.
>
>
> A more exhaustive hyperparameter sweep would likely improve full-FT performance to some extent, but we see our current results as representative of a reasonable, literature-aligned full-FT baseline.
>
>
> `W3 & Q2:  Evaluate on 40k subset of MetaMathQA. “Can the authors perform the experiments on arithmetic reasoning with a larger dataset, e.g., 40k subset of MetaMathQA?”`
>
> Thank you for the suggestion.
>
> We conduct an additional experiment using a 40k MetaMathQA subset on Mistral-7B and Gemma-2 9B, and we observe the same results: ABBA continues to outperform all other PEFT baselines at matched parameter budgets. This suggests that ABBA’s improvements are not specific to the 20k setting and remain consistent as the dataset size increases.
>
> $$
> \\begin{array}{|l|cc|cc|}
> \\hline
> \\textbf{Method} &
> \\begin{array}{c}\\textbf{Mistral-7B}\\\\\\textbf{GSM8K}\\end{array} &
> \\begin{array}{c}\\textbf{Mistral-7B}\\\\\\textbf{MATH}\\end{array} &
> \\begin{array}{c}\\textbf{Gemma-2\\,9B}\\\\\\textbf{GSM8K}\\end{array} &
> \\begin{array}{c}\\textbf{Gemma-2\\,9B}\\\\\\textbf{MATH}\\end{array}\\\\
> \\hline
> \\text{Full FT}   & 66.28 & 18.34 & 79.89 & 39.44\\\\
> \\text{LoRA (r=32)}  & 62.89 & 16.45 & 77.02 & 37.05\\\\
> \\text{DoRA (r=32)}  & 63.24 & 16.67 & 77.46 & 37.40\\\\
> \\text{HiRA (r=32)}  & 64.54 & 17.88 & 78.90 & 38.53\\\\
> \\text{ABBA (r=16)} & 66.36 & 18.51 & 79.93 & 39.27 \\\\
> \\text{ABBA (r=32)} & \\mathbf{67.04} & \\mathbf{18.76} & \\mathbf{80.31} & \\mathbf{39.92}\\\\
> \\hline
> \\end{array}
> $$
>
> We appreciate the reviewer’s thoughtful comments, which have allowed us to strengthen our work. If the points addressed in the rebuttal satisfy your concerns, we would be grateful if you could reflect this in the score. Please let us know if anything else needs clarification.
>
> ---
>
> Thank you again for the constructive suggestions - they have helped us improve our claims. If our responses adequately address your concerns, we kindly ask you to **consider improving the score**. We are happy to clarify any further questions you might have!

---

> > ### Author Response · Authors · 2025-11-26
> >
> > This is a gentle reminder regarding our rebuttal. We know the review process can be demanding and apologize if you were already intending to revisit this review soon. We truly appreciate the attention you have given our submission, and we hope our replies have resolved your concerns. Thank you once more for your careful review, which helped us refine our claims. Please feel free to reach out with any remaining questions!

---

> > ### Comment · Reviewer_FALt · 2025-11-27
> >
> > Thank you for providing a thorough rebuttal. Most of my concerns have been addressed. As I suspected, the performance of full fine-tuning is muh closer to ABBA with the MetaMathQA 40k subset and much better than other PEFT methods (which is also expected). I'm still skeptical that full fine-tuning lags behind by such a large margin on the 20k subset, but given the overhead it might not be worth tuning the hyperparameters in practical settings with small datasets.
> >
> > I would ask the authors to include the results with the 40k subset in the main manuscript to show that larger datasets close the gap between full fine-tuning and ABBA/any PEFT methods, since in larger data regimes (e.g., > 40k or 50k samples), it might be better to use full fine-tuning instead of PEFT, and the authors point this out.
> >
> > I will raise my score to 8.

---

> ### Author Response · Authors · 2025-11-27
>
> Thank you for the thoughtful discussion and for acknowledging that most of your concerns have been addressed.
>
> We agree with your observation that the performance gap between full fine-tuning and ABBA narrows as the dataset size increases. Following your recommendation, we have added the results from fine-tuning on the 40K MetaMathQA subset to the updated main paper (see **Table 6 in Section 4.5**). We also explicitly note that larger datasets tend to reduce the gap between full fine-tuning and ABBA (or other PEFT methods), which might make full fine-tuning a more practical choice in such settings (provided the compute and memory constraints can be met).
>
> Thank you again for your time, and we sincerely appreciate your decision to raise your score in light of our rebuttal.

---

### Official Review · Reviewer_Cd6v · 2025-10-31

**Soundness:** 3
**Presentation:** 3
**Contribution:** 2
**Rating:** 2
**Confidence:** 5

**Summary:**

The paper proposes ABBA, a PEFT adapter that models updates as a Hadamard product of two independently learnable low-rank matrices and provides a Khatri–Rao reformulation so the update can be applied like LoRA without materializing full matrices. Claimed benefits are higher expressivity at a fixed parameter budget and LoRA-like efficiency. The proposed method is tested on commonsense (8 datasets) and arithmetic (GSM8K/MATH) in comparison with rsLoRA, PiSSA, DoRA, LoRA-Pro and HiRA with promising performance obtained.

**Strengths:**

+ The proposed method extends HiRA by replacing its fixed modulation using a learnable factor decouples the updated from $W_0$, which is a technical improvement. The Khatri-Rao factorization is a nice implementation detail.

+ The proposed method is shown to have higher expressivity against LORA via matrix-reconstruction experiments and strong accuracy on commonsense/arithmetic.

**Weaknesses:**

- The main concern is the novelty of this paper is incremental relative to HiRA/MoRA/ReLORA/KronA. The core architectural change is to learn both factors in the Hadamard product instead of tying one to $W_0$ (HiRA). While useful, this feels like a straightforward extension in the space of multiplicative/structured adapters already explored (HiRA, MoRA, KronA, ReLoRA), and the paper’s Related Work acknowledges much of this trajectory. The new factorization is an implementation convenience rather than a substantially new theoretical primitive.

- The proposed method is evaluated only on two tasks: commonsense (8 datasets) and arithmetic (GSM8K/MATH). No instruction-tuning, long-context, code, multilingual, or domain-shift tests—settings where higher-rank adapters might matter most, especially considering the proposed method is claimed to be better than the widespread of LORA.

- Compared with the built-in multi-faceted efficiency of LORA,  the efficiency evdience of the proposed method is not sufficiently compared. Figure 4 reports peak training memory at batch size = 1 and seq len = 256; settings are narrow and may not reflect real training regimes (e.g., larger batches/sequences, activation checkpointing, gradient accumulation). Tables show parameter counts matched to LoRA, sometimes halved (r=16), but parameter count alone is not the whole efficiency story; missing are optimizer-state bytes, activation footprints at realistic batch/seq, and per-step throughput. For example, all experiments run on a single A6000 (48 GB); conclusions about scalability and efficiency may not carry to multi-GPU or larger models.

- LoRA variants are included (rsLoRA, PiSSA, DoRA, LoRA-Pro, HiRA), but QLoRA—a common efficiency baseline combining quantization and adapters—is discussed only in related work and not compared in experiments, and some recent work such as WeGeFT (C. Savadikar et al ICML25) are not discussed and compared.

- Based on the the Khatri–Rao theorem, the proposed method can be treated as a special case of LORA with $B=B_1\odot_r B_2$ and $A=(A_1^\top\odot_r A_2^\top)^\top$, i.e., introducing structures for B and A in LORA, and the proposed initialization of B_1 and A_1. In terms of this, LORA-One (ICML'25) and LORA-GA should be compared.

- Although there are ablations for $\alpha, \{r_1,r_2\}$, and placement, the “adapter chains” variant underperforms, hinting at optimization fragility when stacking. More analysis of failure modes would help (e.g., gradient scales, curvature, stability vs. rank).

**Questions:**

please consider to address the questions in the weaknesses.

---

> ### Author Response · Authors · 2025-11-20
>
> We appreciate the reviewer’s detailed feedback. We address all comments and include **new experiments** to reinforce our claims.
>
> `W1: Concern regarding novelty. “The main concern is the novelty of this paper is incremental relative to HiRA/MoRA/ReLORA/KronA. The core architectural change is to learn both factors in the Hadamard product instead of tying one to W_0 (HiRA)...”`
>
>
> We respectfully disagree that the contribution is merely incremental to HiRA/MoRA/ReLoRA/KronA. Our main contribution is a *new parameterization class* for PEFT adapters: updates of the form
> $$
> \Delta W = (B_1A_1)\odot(B_2A_2),
> $$
> i.e., the Hadamard product of **two independently learned low-rank matrices**, together with a Khatri–Rao reformulation that makes this class trainable with LoRA-like memory and compute.
>
> This parameterization is explicitly motivated by the complementary shortcomings of LoRA and HiRA: (i) LoRA can only express rank-$r$ updates, which is empirically insufficient for many layers; (ii) HiRA can in principle express higher-rank updates, but its update is *tied elementwise to $W_0$*, which severely restricts the update space whenever the oracle update is not a simple modulation of $W_0$ (we show such cases in our reconstruction experiments). ABBA sits in the “middle ground”: it decouples from $W_0$ like LoRA, but recovers HiRA’s multiplicative flavour and can achieve rank up to $r^2$ while keeping the same parameter budget.
>
> The “natural extension” criticism would be valid if any generic multiplicative reparameterization were both expressive and efficient, but this is exactly where ABBA is non-trivial. Naively combining two parameter-efficient matrices multiplicatively leads to an intermediate high rank object.
>
> Our use of the *specific* algebraic property of the Hadamard product and the associated Khatri–Rao identity is what makes this parameterization practically viable: we obtain a strictly larger function class without paying the cost of explicitly handling rank-$r^2$ matrices. Other structured adapters you mention (MoRA, KronA, ReLoRA) either (a) remain tied to $W_0$ or other fixed structure, or (b) increase compute/memory substantially when targeting higher rank. None of them, to our knowledge, learns two fully decoupled low-rank factors and exploits the Khatri–Rao structure to obtain high-rank behaviour under a LoRA-like budget.
>
>
> `W2: Evaluate on other tasks: “The proposed method is evaluated ….”`
>
> Thanks for the suggestion.
>
> We now evaluate ABBA in comparison with other PEFT methods on code generation tasks. Specifically, we fine-tuned Llama-3.2-1B on a 100K subset of the CodeFeedback dataset and evaluated performance using Pass@1 on HumanEval. As shown in the results below, ABBA again performs very strongly, outperforming all other PEFT variants and even full fine-tuning (all PEFT methods are run at rank 32). This indicates that the benefits of our method generalize to more challenging tasks such as code generation.
>
> We include these results in **Table 10 (Appendix H)** of the updated paper.
>
>
> $$
> \\begin{array}{|l|c|c|}
> \\hline
> \\textbf{Method} &
> \\begin{array}{c}\\textbf{\\# Params}\\end{array} &
> \\begin{array}{c}\\textbf{HumanEval}\\;(\uparrow)\\end{array} \\\\
> \\hline
> \\text{Full FT}   & 1.24\\text{B}  & 23.17 \\\\
> \\text{LoRA}      & 22.54\\text{M} & 20.73 \\\\
> \\text{HiRA}      & 22.54\\text{M} & 21.95 \\\\
> \\rowcolor{cyan!10}\\text{ABBA} & 22.54\text{M} & \mathbf{25.61} \\\\
> \\hline
> \\end{array}
> $$

---

> ### Author Response · Authors · 2025-11-20
>
> `W3: Clarifications around efficiency: “Compared with the built-in multi-faceted efficiency of LoRA….”`
>
> We now report detailed memory and timing measurements under the suggested training settings, using 8 gradient-accumulation steps, activation checkpointing, batch size 4, and sequence length 512. We find that ABBA closely matches LoRA across all efficiency metrics in realistic settings.
>
> **Space (GPU memory)**
>
>  We observe that peak GPU memory consumption for ABBA and LoRA remains nearly identical up to (r=128). These results are shown in **Figure 9 (Appendix L)** of the updated paper.
>
> **Timing (latency / throughput)**
>
>  Training-time metrics, latency per step and tokens/sec, also remain very similar between ABBA and LoRA up to (r=128). These results are shown in **Figure 7 and 8 (Appendix K)** in the updated manuscript.
>
> ---
>
> Due to hardware constraints (a single 48GB A6000 GPU), we are unable to run fine-tuning experiments on models larger than 10B parameters. However, the relative memory efficiency of ABBA versus LoRA does not depend on the GPU used, and the timing behavior scales proportionally with model size in both methods. Thus, while we cannot report larger-model experiments, the comparative efficiency trends we observe are expected to hold more broadly.
>
> ---
>
> Finally, we emphasize that our experimental scale and settings are consistent with prior PEFT work, including HiRA, MoRA, rsLoRA, LoRA-GA, and LoRA-Pro, which also evaluate models in the 7B–9B range under single-GPU training. We therefore believe that our analysis provides a fair and representative assessment of ABBA’s efficiency with respect to the broader PEFT literature.
>
> `W6: “Although there are ablations for $\alpha, {r_1,r_2}$, and placement, the “adapter chains” variant underperforms, hinting at optimization fragility when stacking. More analysis of failure modes would help (e.g., gradient scales, curvature, stability vs. rank).”`
>
> Thanks for the suggestion.
>
> In the updated paper, we provide additional analysis of the “adapter chains” variant by reporting gradient-norm behavior for the chained ABBA setup (four adapter pairs) versus the standard ABBA configuration (two pairs). These results are shown in **Figure 6** (Appendix J) in the updated paper.
>
> Consistent with the discussion in Section 4.5, the chained variant exhibits substantially higher gradient-norm instability, suggesting that stacking multiple ABBA blocks amplifies optimization sensitivity. This analysis clarifies the failure mode and explains the slight performance degradation observed with the chained design.
>
> ---
>
>
> `W4: LoRA variants are included (rsLoRA, PiSSA, DoRA, LoRA-Pro, HiRA), but QLoRA-a common efficiency baseline combining quantization and adapters-is discussed only in related work and not compared in experiments, and some recent work such as WeGeFT are not discussed and compared.`
>
>
> We respectfully note that QLoRA and WeGeFT address *orthogonal* aspects of the fine-tuning problem and therefore are not natural baselines when evaluating a new **adapter parameterization**.
>
>
> QLoRA quantizes the *base model weights* to reduce memory footprint; it is fully complementary to LoRA-style adapters and can be applied on top of **any** adapter architecture, including ABBA. In other words, QLoRA is not a competing parameterization but a system-level optimization compatible with ABBA in exactly the same way it is with LoRA, DoRA, or HiRA. Moreover, it is well known that while QLoRA is highly parameter- and memory-efficient, it does not exhibit higher downstream performance than full-precision LoRA due to quantization-induced loss. Because QLoRA generally underperforms its full-precision LoRA counterpart, it would not serve as a stronger or more informative baseline than LoRA itself for evaluating adapter **parameterizations**.
> For fairness, we therefore compare against methods that modify the *update rule itself* (e.g., LoRA, HiRA, DoRA, LoRA-Pro, PiSSA), which is precisely the class of approaches to which ABBA belongs.
>
>
> ---
>
> Regarding WeGeFT, it solves a fundamentally different problem from ABBA. WeGeFT alters the weight-generation mechanism for full fine-tuning by producing updated weights through learned transformations of the base model parameters. We have added a discussion of WeGeFT in the Related Work section (Lines 756–757). It does not target low-rank expressivity, rank-deficiency, or alternative adapter parameterizations. While the PEFT landscape is broad, it is neither practical nor standard to compare against methods that act on different parts of the fine-tuning pipeline.
>
> We therefore focus on the most widely used and directly comparable structured-low-rank parameterizations. We believe this set (LoRA, HiRA, DoRA, LoRA-Pro, PiSSA, rsLoRA) provides a clear and sufficiently comprehensive baseline suite for situating ABBA among existing adapter architectures.

---

> > ### Author Response · Authors · 2025-11-20
> >
> > `W5: Based on the Khatri–Rao theorem, the proposed method can be treated as a special case of LoRA….., and the proposed initialization of B_1 and A_1. In terms of this, LORA-One (ICML'25) and LORA-GA should be compared.`
> >
> >
> > We respectfully disagree with this reduction. While the two approaches can be equivalent at the level of the *resulting update matrix*, they are **not** equivalent in terms of the *parameterization* or the *optimization dynamics*, which are central to ABBA’s behavior and performance.
> >
> > In precisely the same sense, LoRA itself can be viewed as a “special case” of full fine-tuning, because full FT can represent any rank-$r$ update; yet LoRA is regarded as a distinct method because its reparameterization fundamentally changes the learning dynamics, conditioning, and gradient flow.
> >
> > The same is true here: ABBA’s update belongs to the function class of rank-$r^2$ LoRA matrices, but our factorization imposes a structured constraint on $B$ and $A$ that **cannot be reduced to standard LoRA without destroying the reparameterization**. This structure changes (i) how gradients propagate through each factor, (ii) the curvature properties seen by the optimizer, and (iii) how effective rank grows during training.
> >
> > Since the optimization in parameter space, not merely the representational class, determines empirical performance, ABBA behaves very differently from LoRA with rank $r^2$. Our experiments confirm this: the two methods yield different learned matrices, different reconstruction paths, and different downstream performance. Thus, the relationship is analogous to LoRA vs. full FT: the representable set may overlap, but the parameterization constitutes a distinct and practically meaningful method rather than a trivial special case.
> >
> > ---
> >
> > For this reason, we do not consider LoRA-One or LoRA-GA to be the most relevant baselines. These methods modify *initialization or optimization heuristics* within the standard LoRA parameterization, whereas ABBA proposes a **new parameterization class** altogether. We already compare against PiSSA, which is the most prominent initialization-based variant and directly targets improved conditioning. We have added a discussion of LoRA-One in the Related Work section (Lines 754–755).
> >
> > Nonetheless, to address the reviewer’s concern, we additionally include comparisons with LoRA-GA on the arithmetic reasoning benchmarks in our updated results below. We can see that ABBA consistently outperforms LoRA-GA for all tasks and models.
> >
> > $$ \\begin{array}{|l|cc|cc|}
> > \\hline
> > \\textbf{Method} & \\begin{array}{c}\\textbf{Mistral-7B}\\\\\\textbf{GSM8K}\\end{array} &
> > \\begin{array}{c}\\textbf{Mistral-7B}\\\\\\textbf{MATH}\\end{array} & \\begin{array}{c}\\textbf{Gemma-2\\,9B}\\\\\\textbf{GSM8K}\\end{array} & \\begin{array}{c}\\textbf{Gemma-2\\,9B}\\\\\\textbf{MATH}\\end{array}\\\\
> > \\hline
> > \\text{LoRA (r=32)} & 61.94 & 15.98 & 76.19 & 36.56\\\\
> > \\text{LoRA-GA (r=32)}  & 62.87 & 16.66 & 77.28 & 37.13\\\\
> > \\text{ABBA (r=16)} & 64.97 & 18.06 & 78.70 & 38.41\\\\
> > \\text{ABBA (r=32)} & \\mathbf{66.26} & \\mathbf{18.08} & \\mathbf{79.76} & \\mathbf{39.18}
> > \\\\
> >  \\hline
> > \\end{array} $$
> >
> > ---
> >
> >
> >
> > We thank the reviewer again for their review. If the points addressed in the rebuttal satisfy your concerns, we kindly ask that you **consider increasing the score**. Please let us know if anything else would benefit from clarification.

---

> > > ### Author Response · Authors · 2025-11-26
> > >
> > > This is a gentle reminder regarding our rebuttal. We fully understand the reviewers’ workload and apologize if you were already planning to look at it soon. We hope our responses have addressed your concerns thoroughly. Thank you again for your review. If anything remains unclear, we would be glad to clarify!

---

> > > > ### Comment · Reviewer_Cd6v · 2025-11-27
> > > > **Thank you for the rebuttal**
> > > >
> > > > I thank the authors for their rebuttal, and have also read other reviewers' comments.
> > > >
> > > > Overall, the proposed ABBA ($\Delta W=(B_1A_1)\odot (B_2A_2)$) still looks structurally similar to HiRA ($\Delta W=(BA)\odot W_0$). Even though ABBA drops the dependence on $W_0$, ABBA is still a multiplicative LoRA variant, and the jump from one multiplicative factor (in HiRA) to two seem like a natural extension, not a fundamentally new conceptual framework.  The rebuttal explains why it's different, but does **not provide a formal characterization of what ABBA can express that HiRA categorically can not** (e.g. showing strict containment of LoRA and HiRA within ABBA, or a proof of irreducibility in terms of HiRA can not simulate ABBA variants).  In the meanwhile, based on the motivation and observation in WeGeFT,  dependency on $W_0$ could be a good leverage since PEFT is to exploit the prior knowledge encoded in the pretrained model while exploring necessary information in downstream tasks. This might be useful, especially when more and more powerful pretrained base models will be available. Furthermore,  if we do PEFT from instruct models, conditioning on pretrained weights may help preserve better safety alignment.
> > > >
> > > > I totally understood the fact that computational resources are limited, so the experiments were run using a single A6000 GPU.   But, this does not prevent me from questioning the scalability in practice. Especially, for large models, pretrained weights have encoded even more prior knowledge, would HiRA be actually better due to the dependency?
> > > >
> > > > Consider the efforts in the rebuttal, I think the overall novelty claim remains largely theoretical, not verified at the scale where it matters. I will slightly increase my score, but overall lean to weak reject based on the reasons stated above.

---

> ### Author Response · Authors · 2025-11-27
>
> Thank you for your thoughtful response.
>
> We fully appreciate the reviewer’s perspective. While it is not possible to provide a theoretical proof that ABBA is strictly more expressive than HiRA, we demonstrate this empirically across the models evaluated in our paper. We believe these results highlight the strengths of our approach.
>
> We also understand the reviewer’s concerns regarding scalability. Unfortunately, a formal demonstration of scalability is outside the scope of what we can provide at this stage. Nevertheless, we really value the constructive feedback and the insightful discussion throughout the review process.
>
> We are grateful for the reviewer’s engagement and appreciate the decision to raise the score.

---

### Official Review · Reviewer_XkLT · 2025-10-31

**Soundness:** 3
**Presentation:** 2
**Contribution:** 3
**Rating:** 6
**Confidence:** 4

**Summary:**

This paper proposes ABBA, a new parameter-efficient fine-tuning (PEFT) method that models the weight update as the Hadamard product of two independently learnable low-rank matrices. Unlike HiRA, which ties the update to the pretrained weights $W_0$, ABBA fully decouples both components, allowing them to be optimized freely. The authors further derive an efficient Khatri–Rao factorization so that ABBA can be implemented with the same memory and compute efficiency as LoRA while offering much higher expressivity. Extensive experiments on Llama-3.2 (1B & 3B), Mistral-7B, and Gemma-2 9B demonstrate consistent improvements over LoRA, HiRA, DoRA, and other PEFT baselines on commonsense and arithmetic reasoning benchmarks.

**Strengths:**

1. The idea of composing two learnable low-rank modules via a Hadamard product is conceptually clean and represents a clear generalization of LoRA and HiRA. The method increases effective rank while maintaining strict parameter efficiency.
2. The paper introduces Khatri–Rao reformulation and a well-motivated rank-stability theorem, which explains how scaling should depend on $r_1,r_2$.
3. Results across four foundation models and multiple reasoning tasks show large and consistent gains. The authors carefully control for parameter count and initialization, making comparisons fair.

**Weaknesses:**

1. The paper proves a scaling law for stability but does not empirically show how optimization behaves under varying $r_1, r_2$ or initialization errors. Gradient norm or loss-landscape visualizations would strengthen claims about stable training.
2. It remains unclear whether the performance gain comes from the Hadamard structure itself or simply from doubling the number of learnable matrices. An ablation removing the Hadamard product (e.g., summation or concatenation) would clarify this.
3. While Section 2.4 claims that ABBA is more expressive than LoRA, the analysis relies primarily on empirical reconstruction errors. It will be better that if authors can provide formal proof or analysis showing that ABBA’s representational space strictly contains LoRA’s or HiRA’s.

**Questions:**

Please refer Weaknesses

---

> ### Author Response · Authors · 2025-11-20
>
> We thank the reviewer for the constructive feedback. Below, we address each of the points raised, and run **additional experiments** to strengthen our claims.
>
> `W1: “The paper proves a scaling law for stability but does not empirically show how optimization behaves under varying r1,r2 or initialization errors. Gradient norm or loss-landscape visualizations would strengthen claims about stable training.”`
>
> Thank you for this helpful suggestion, we agree that adding empirical evidence strengthens our claims here.
> In addition to the theoretical result in Section 2.3 (Theorem 2), we have now included gradient-norm plots comparing several scaling strategies:
>
>  $s_{\text{ABBA}} \in \Theta(1),\quad
>  \Theta\left(\tfrac{1}{(r_1 r_2)^{1/6}}\right),\quad
>  \Theta\left(\tfrac{1}{(r_1 r_2)^{1/4}}\right),\quad
>  \Theta\left(\tfrac{1}{(r_1 r_2)^{1/2}}\right).
>  $
>
>  As shown in **Figure 5 (Appendix I)** of the updated paper, the scaling derived from theory $s_{\text{ABBA}} \in \Theta\left(\tfrac{1}{\sqrt{r_1 r_2}}\right)$ consistently produces the most stable gradient norms, in line with our theoretical findings. These results provide additional empirical support for the stability claims presented in the paper.
>
>
> `W2: “It remains unclear whether the performance gain comes from the Hadamard structure itself or simply from doubling the number of learnable matrices. An ablation removing the Hadamard product (e.g., summation or concatenation) would clarify this.”`
>
> Thank you for suggesting this ablation, it provides a valuable clarification of where the performance gains originate.
> We performed the proposed experiment by replacing the Hadamard product with a summation of two rank-(r/2) LoRA adapters:
>  $$
>  \ \Delta W = B_1 A_1 + B_2 A_2.
>  $$
>
> Crucially, this construction is functionally equivalent to a single rank-(r) LoRA adapter, since any rank ≤r matrix can be written as a sum of two rank ≤r/2 matrices (e.g., by splitting the truncated SVD). Thus, this ablation matches the representational capacity of the rank-(r) LoRA baseline already included in our experiments. We refer to this variant as LoRA-Sum.
>
> The results are shown below:
>
> $$ \\begin{array}{|l|cc|cc|}
> \\hline
> \\textbf{Method} & \\begin{array}{c}\\textbf{Mistral-7B}\\\\\\textbf{GSM8K}\\end{array} &
> \\begin{array}{c}\\textbf{Mistral-7B}\\\\\\textbf{MATH}\\end{array} & \\begin{array}{c}\\textbf{Gemma-2\\,9B}\\\\\\textbf{GSM8K}\\end{array} & \\begin{array}{c}\\textbf{Gemma-2\\,9B}\\\\\\textbf{MATH}\\end{array}\\\\
> \\hline
> \\text{LoRA (r=32)} & 61.94 & 15.98 & 76.19 & 36.56\\\\
> \\text{LoRA-Sum (r=32)} & 62.03 & 15.93 & 76.21 & 36.45\\\\
> \\text{ABBA (r=16)} & 64.97 & 18.06 & 78.70 & 38.41\\\\
> \\text{ABBA (r=32)} & \\mathbf{66.26} & \\mathbf{18.08} & \\mathbf{79.76} & \\mathbf{39.18}
> \\\\
>  \\hline
> \\end{array} $$
>
> As shown above, LoRA-Sum performs nearly identically to standard LoRA, indicating that the additional matrices alone do not account for the performance gains. The improvements therefore stem from the Hadamard structure introduced by ABBA, which substantially outperforms both LoRA and LoRA-Sum.

---

> > ### Author Response · Authors · 2025-11-20
> >
> > `W3: “The expressivity analysis relies on empirical reconstruction. A more formal statement that ABBA’s representational space strictly contains LoRA/HiRA would improve the claim."`
> >
> > We thank the reviewer for this helpful suggestion. We clarify the theoretical picture below.
> >
> > ---
> >
> > ### **Why ABBA cannot admit an SVD-style closed-form optimum**
> >
> > Unlike the classical low-rank problem
> >
> > $$
> > \min_{\text{rank}(X)\le r} \|M - X\|_F^2,
> > $$
> >
> > where the Eckart–Young theorem provides an optimal closed-form solution via SVD, the ABBA decomposition
> >
> > $$
> > M \approx (B_1A_1)\odot(B_2A_2)
> > $$
> >
> > leads to a **quartic, non-convex optimization problem**. Prior work (Ciaperoni et al., 2024; Wertz et al., 2025) explicitly shows that *no closed-form optimal solution is known* for Hadamard-structured factorizations, and practical methods rely on numerical optimization. Our approach follows this established practice.
> >
> > ---
> >
> > ### **Error bounds via SVD-based reconstruction**
> >
> > While ABBA lacks a closed-form optimum, one can still derive **SVD-based upper and lower bounds**:
> >
> > - ABBA can represent any rank-$r$ matrix trivially (choose $B_2A_2 = \mathbf{1}$ and let $B_1A_1$ be rank-$r$).
> > - In the best case, ABBA can represent matrices of rank up to **$r^2$** (depending on alignment of the factors).
> >
> > Therefore, the reconstruction error of ABBA is bounded between the optimal errors of rank-$r$ and rank-$r^2$ SVD approximations:
> >
> > $$
> > \sum_{i=r+1}^{\min(m,n)} \sigma_{i}^2 \geq\ \min_{ABBA} \|M - \Delta W\|_F^2
> > $$
> >
> > and
> >
> > $$
> > \sum_{i=r^2+1}^{\min(m,n)} \sigma_{i}^2 \le\ \min_{ABBA} \|M - \Delta W\|_F^2
> > $$
> >
> >
> > where $\sigma_i$ are the singular values of $M$.
> > This bound illustrates that **ABBA’s approximation quality depends on the singular value decay** of the target matrix.
> >
> > ---
> >
> > ### **Empirical bounds are much tighter in practice**
> >
> > Although the theoretical bounds above may be loose (as with many structured decompositions), **empirical results from prior work** (Ciaperoni et al., 2024; Wertz et al., 2025) show that Hadamard/Khatri–Rao–based decompositions consistently achieve **far smaller reconstruction errors than predicted by worst-case theory**, often outperforming low-rank SVD under equal parameter budgets.
> >
> > Our own experiments follow this methodology and show ABBA outperforming LoRA across all matrix classes tested.
> >
> > ---
> >
> > We hope these clarifications address the reviewer’s request for stronger formal grounding while remaining faithful to what is theoretically known about Hadamard-structured factorizations.
> >
> > ---
> >
> > ### **References**
> >
> > - Ciaperoni, M., Gionis, A., & Mannila, H. (2024). *The Hadamard decomposition problem*. Data Mining and Knowledge Discovery, 38(4), 2306–2347.
> > - Wertz, S., Vandaele, A., & Gillis, N. (2025). *Efficient algorithms for the Hadamard decomposition*. arXiv:2504.13633.
> > ---
> >
> >
> > We thank the reviewer again for the helpful suggestions, which have helped us strengthen our claims. If our responses adequately address your concerns, we kindly ask you to **consider increasing your score**. We are happy to clarify any remaining points you might have!

---

> > > ### Author Response · Authors · 2025-11-26
> > >
> > > This is a gentle reminder regarding our rebuttal. We fully understand the reviewers’ workload and apologize if you were already planning to look at it soon. We truly appreciate the time you have already spent reviewing our work. We hope our responses have addressed your concerns as thoroughly as possible. Thank you again for the careful and thoughtful review, which helped us strengthen our claims. We are happy to clarify any remaining questions you might have!

---

> > > > ### Comment · Reviewer_XkLT · 2025-11-26
> > > >
> > > > I appreciate the authors' comprehensive and detailed response. Most of my concerns have been addressed. However, building upon other reviewers, I believe the paper would be significantly strengthened if the authors could provide further theoretical or experimental analysis to validate that ABBA represents a more generalized case compared to existing methods such as HiRA, MoRA, ReLORA, and KronA.

---

> ### Author Response · Authors · 2025-11-27
>
> Thank you for acknowledging that our responses addressed your concerns, and thanks for your suggestions.
>
>
> In response to your requests, we have added new comparisons of ABBA with high rank baselines, namely **MoRA**, **KronA**, and **Flora** (as also requested by reviewer Z4En), in addition to **HiRA**. Due to time constraints, we are not able to run experiments on all papers mentioned by the reviewer. We also note that ReLoRA requires full fine tuning for 25 percent of the training time, which leads to memory issues and is more commonly used for efficient pretraining.
>
>
>
>
> For these experiments, we fine tuned Llama 3.1 8B on a 100K subset of the CodeFeedback dataset and evaluated performance using Pass@1 on HumanEval. We present results comparing ABBA with LoRA, HiRA, Flora, MoRA, and KronA in the table below. The results are very promising and show that **ABBA continues to outperform the additional high rank baselines**, along with the baselines already evaluated in the paper.
>
>
>
>
> $$
> \\begin{array}{|l|c|c|}
> \\hline
> \\textbf{Method} &
> \\begin{array}{c}\\textbf{HumanEval}\\;(\uparrow)\\end{array} \\\\
> \\hline
> \\text{Full FT}   & 48.54 \\\\
> \\text{LoRA}  & 45.73 \\\\
> \\text{KronA}  & 45.12 \\\\
> \\text{Flora} & 45.56 \\\\
> \\text{MoRA}   & 47.28 \\\\
> \\text{HiRA}    & 48.17 \\\\
> \\text{ABBA} & \mathbf{49.39} \\\\
> \\hline
> \\end{array}
> $$
>
>
> ---
>
>
> We believe that this new comparison further supports ABBA’s position as a very strong high-rank PEFT method. We kindly ask the reviewer to **consider increasing their score** if they feel that our responses have clarified and strengthened the claims in the paper. Thank you again for your time.

---

> > ### Author Response · Authors · 2025-12-02
> >
> > Thank you again for your review and discussions! We have included comparisons with the additional high-rank baselines in **Table 12 (Appendix M)**.

---

### Author Response · Authors · 2025-12-02
**Summary of discussion period (avg. score increased to 6.5 from 4.5)**

This is our attempt to summarize the discussion period as openly and honestly as we can. All of the points below can be verified directly from the discussion record.

---

`Reviewer FALt` wrote that “*Most of my concerns have been addressed*” and **raised their score from 6 to 8**, stating clearly, “*I will raise my score to 8.*”

`Reviewer Z4En` wrote that “*My concerns have been adequately addressed. I'm raising my score to 8,*” thereby **raising their score from 4 to 8**.

`Reviewer Cd6v` **increased their score from 2 to 4**, as noted in their comment, “*I will slightly increase my score.*”

`Reviewer XkLT`, who originally gave a 6, stated that “*Most of my concerns have been addressed.*” They also mentioned in a follow-up discussion that “*I believe the paper would be significantly strengthened*” with comparisons to high-rank baselines like HiRA, MoRA, and KronA. We included those comparisons in our response, and we believe there was a strong likelihood they would have raised their score to 8 if the discussion had not ended abruptly.

These updates **improved our average to 6.5 from 4.5** before the scores were reverted, and potentially to 7 if the discussion had proceeded without interruption.

---

We hope this context clarifies that our detailed rebuttal addressed the key concerns of all reviewers thoroughly and satisfactorily. We request that the AC consider these factors when evaluating our paper in these unexpected scenarios. We remain grateful for the time and effort invested in reviewing our work.

---

> ### Author Response · Authors · 2025-12-02
> **Summary of major experiments and revisions**
>
> Below is a brief summary of the major experiments and revisions we added during the discussion period in response to reviewer feedback.
>
> ---
>
> `Performance on complex tasks like coding (Reviewers Cd6v, Z4En)`
>
> We add fine-tuning results for Llama-3.2 1B and Llama-3.1 8B on code generation (**Appendix H, Table 11** and **Appendix M, Table 12**). ABBA consistently outperforms all methods, even on complex code generation tasks.
>
> `Comparison with high-rank baselines: Flora, MoRA, and KronA (Reviewers XkLT, Z4En)`
>
> We include comparisons with these additional high-rank baselines (**Appendix M, Table 12**). ABBA consistently outperforms all high-rank methods, in addition to HiRA and other baselines evaluated originally.
>
> `Detailed comparison of computational cost of ABBA with LoRA (Reviewers Cd6v, Z4En)`
>
> We show that peak memory usage and training time for ABBA and LoRA remain similar under all settings, including at high ranks (**Appendix K, Figures 7-8** and **Appendix L, Figure 9**).
>
> `Performance on a larger dataset like the 40k subset of MetaMathQA (Reviewer FALt)`
>
> ABBA continues to outperform other methods even on this larger data subset (**Section 4.5, Table 6**).
>
> `Gradient norm plots for different scaling strategies and adapter chains (Reviewers XkLT, Cd6v)`
>
> We show that our scaling strategy produces the most stable gradient norms, consistent with our theoretical findings (**Appendix I, Figure 5**). The chained variant shows greater instability than the standard ABBA model (**Appendix J, Figure 6**).
>
> `Clarifying performance of full FT vs ABBA (Reviewer FALt)`
>
> We add a discussion clarifying how we expect full FT to perform relative to ABBA (**Section 4.5**).
>
> `Additional intuition behind why the ABBA representational space is more expressive than LoRA/HiRA (Reviewers XkLT, Z4En)`
>
> We provide additional intuition in our responses to Reviewers XkLT and Z4En.
>
> ---
>
> We believe these updates meaningfully addressed all reviewer concerns and helped clarify and strengthen our paper's claims.

---

### Meta-Review · Area_Chair_rVkE · 2025-12-19

**Summary:**

The article introduces ABBA, a novel architecture for Parameter-Efficient Fine-Tuning (PEFT) of foundation models. Addressing the expressivity limitations of Low-Rank Adaptation (LoRA) and the structural constraints of High-Rank Adaptation (HiRA), the authors propose modeling weight updates as the Hadamard product of two independently learnable low-rank matrices. This approach decouples the update from pre-trained weights, enabling high-rank adaptation while maintaining parameter efficiency through a mathematically exact Khatri–Rao factorization. Empirical evaluations across Llama, Mistral, and Gemma models demonstrate that ABBA achieves state-of-the-art performance on arithmetic and commonsense reasoning benchmarks, consistently outperforming existing methods like LoRA, DoRA, and HiRA.

**Reviewer Concerns:**

During the preliminary review, the submission received a mixed reception, with initial ratings ranging from 2 to 6. Reviewers commended the clear theoretical motivation, the clever use of Khatri-Rao factorization to ensure computational feasibility, and the strong empirical gains on reasoning tasks. However, significant concerns were raised regarding the method's novelty relative to HiRA, the lack of comparison against other high-rank PEFT baselines such as MoRA and KronA, and the absence of evaluation on more complex domains like code generation. Critics also questioned the scalability of the method regarding computational overhead and requested deeper analysis of the optimization stability and gradient behavior.


The authors provided extensive new experiments during the discussion period, including results on code generation tasks where ABBA outperformed all baselines, and direct comparisons with high-rank methods like Flora, MoRA, and KronA, establishing ABBA's superior expressivity. Furthermore, detailed memory and latency analyses clarified that ABBA maintains LoRA-like efficiency, and new gradient norm visualizations validated the proposed scaling laws. Consequently, the majority of reviewers moved to strong acceptance, acknowledging that the revisions comprehensively addressed their concerns regarding novelty, robustness, and baselines. The work offers a valuable, efficient, and highly expressive contribution to the PEFT landscape.

**Reviewer Scores:**

The rebuttal phase was exceptionally productive, leading to a consensus for acceptance. Reviewer FALt raised their score from 6 to 8, and Reviewer Z4En raised their score from 4 to 8, both citing that the new experiments and clarifications fully addressed their concerns. Reviewer XkLT, who initially gave a 6, explicitly stated in the discussion that most concerns were addressed and that they intended to raise their score to 8 following the inclusion of high-rank baselines. Reviewer Cd6v increased their score from 2 to 4, acknowledging the rebuttal efforts but retaining reservations about the theoretical novelty compared to HiRA.

---

### Decision · Program_Chairs · 2026-01-26

Accept (Poster)